# F8Net: Fixed-Point 8-bit Only Multiplication for Network Quantization

**Qing Jin**[1,2]* **Jian Ren**[1] **Richard Zhuang**[1] **Sumant Hanumante**[1] **Zhengang Li**[2]
**Zhiyu Chen**[3] **Yanzhi Wang**[2] **Kaiyuan Yang**[3] **Sergey Tulyakov**[1]
[1]Snap Inc.    [2]Northeastern University, USA    [3] Rice University, USA

## ABSTRACT

Neural network quantization is a promising compression technique to reduce memory footprint and save energy consumption, potentially leading to real-time inference. However, there is a performance gap between quantized and full-precision models. To reduce it, existing quantization approaches require high-precision INT32 or full-precision multiplication during inference for scaling or dequantization. This introduces a noticeable cost in terms of memory, speed, and required energy. To tackle these issues, we present F8Net, a novel quantization framework consisting of only fixed-point 8-bit multiplication. To derive our method, we first discuss the advantages of fixed-point multiplication with different formats of fixed-point numbers and study the statistical behavior of the associated fixed-point numbers. Second, based on the statistical and algorithmic analysis, we apply different fixed-point formats for weights and activations of different layers. We introduce a novel algorithm to automatically determine the right format for each layer during training. Third, we analyze a previous quantization algorithm—parameterized clipping activation (PACT)—and reformulate it using fixed-point arithmetic. Finally, we unify the recently proposed method for quantization fine-tuning and our fixed-point approach to show the potential of our method. We verify F8Net on ImageNet for MobileNet V1/V2 and ResNet18/50. Our approach achieves comparable and better performance, when compared not only to existing quantization techniques with INT32 multiplication or floating-point arithmetic, but also to the full-precision counterparts, achieving state-of-the-art performance.

## 1 INTRODUCTION

Real-time inference on resource-constrained and efficiency-demanding platforms has long been desired and extensively studied in the last decades, resulting in significant improvement on the trade-off between efficiency and accuracy (Han et al., 2015; Liu et al., 2018; Mei et al., 2019; Tanaka et al., 2020; Ma et al., 2020; Mishra et al., 2020; Liang et al., 2021; Jin et al., 2021; Liu et al., 2021). As a model compression technique, quantization is promising compared to other methods, such as network pruning (Tanaka et al., 2020; Li et al., 2021; Ma et al., 2020; 2021a; Yuan et al., 2021) and slimming (Liu et al., 2017; 2018), as it achieves a large compression ratio (Krishnamoorthi, 2018; Nagel et al., 2021) and is computationally beneficial for integer-only hardware. The latter one is especially important because many hardwares (e.g., most brands of DSPs (Ho, 2015; QCOM, 2019)) only support integer or fixed-point arithmetic for accelerated implementation and cannot deploy models with floating-point operations. However, the drop in performance, such as classification accuracy, caused by quantization errors, restricts wide applications of such methods (Zhu et al., 2016).

To address this challenge, many approaches have been proposed, which can be categorized into *simulated* quantization, *integer-only* quantization, and *fixed-point* quantization (Gholami et al., 2021). Fig. 1 shows a comparison between these implementations. For simulated quantization, previous works propose to use trainable clipping-levels (Choi et al., 2018), together with scaling techniques on activations (Jin et al., 2020b) and/or gradients (Esser et al., 2019), to facilitate training for the quantized models. However, some operations in these works, such as batch normalization (BN),

---

*Work done during an internship at Snap Inc. Code is available at https://github.com/snap-research/F8Net.

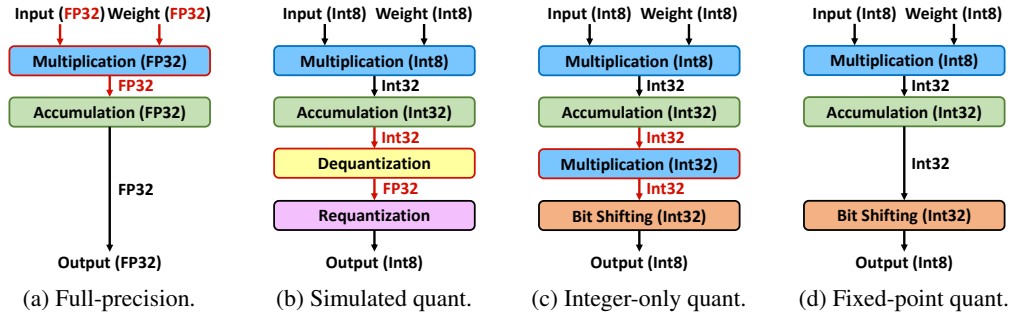

Figure 1: Inspired by Gholami et al. (2021), we show the comparison of full-precision model (presented in (a)) and different quantizations settings: (b) simulated quantization; (c) integer-only quantization; and (d) fixed-point quantization. Note the combination of last two operations in integer-only quantization is termed as dyadic scaling in literature (Yao et al., 2021).

are conducted with full-precision to stabilize training (Jin et al., 2020b; Esser et al., 2019), limiting the practical application of integer-only hardware. Meanwhile, integer-only quantization, where the model inference can be implemented with integer multiplication, addition, and bit shifting, has shown significant progress in recent studies (Jacob et al., 2018; Yao et al., 2021; Kim et al., 2021). Albeit floating-point operations are removed to enable models running on devices with limited support of operation types, INT32 multiplication is still required for these methods. On the other hand, fixed-point quantization, which also applies low-precision logic for arithmetic, does not require INT32 multiplication or integer division. For example, to replace multiplication by bit shifting, Jain et al. (2019) utilize trainable power-of-2 scale factors to quantize the model.

In this work, we adopt fixed-point quantization. Our work differs from previous efforts (Jain et al., 2019) in three major aspects. First, to determine the minimum error quantization threshold, we conduct statistical analysis on fixed-point numbers. Second, we unify parameterized clipping activation (PACT) and fixed-point arithmetic to achieve high performance and high efficiency. Third, we discuss and propose quantization fine-tuning methods for different models. We dub our method as F8Net, as it consists in only **F**ixed-point **8**-bit multiplication employed for **Net**work quantization. We thoroughly study the problem with fixed-point numbers, where only INT8 multiplication is involved, without any INT32 multiplication, neither floating-point nor fixed-point types. Throughout this paper we focus on 8-bit quantization, the most widely supported case for different devices and is typically sufficient for efficiency and performance requirements. Our contribution can be elaborated as follows.

- We show 8-bit fixed-point number is able to represent a wide range of values with negligible relative error, once the format is properly chosen (see Fig. 3 and Fig. 4). This critical characteristic enables fixed-point numbers a much stronger representative capability than integer values.

- We propose a method to determine the fixed-point format, also known as fractional length, for weights and activations using their variance. This is achieved by analyzing the statistical behaviors of fixed-point values of different formats, especially those quantized from random variables with normal distribution of different variances. The analysis reveals the relationship between relative quantization error and variance, which further helps us build an approximated formula to determine the fractional length from the variance.

- We develop a novel training algorithm for fixed-point models by unifying fixed-point quantization and PACT (Choi et al., 2018). Besides, we show the impact of fractional length sharing for residual blocks, which is also important to obtain good performance for quantized models.

- We validate our approach for various models, including MobileNet V1/V2 and ResNet18/50 on ImageNet for image classification, and demonstrate better performance than existing methods that resort to 32-bit multiplication. We also integrate the recent proposed fine-tuning method to train quantized models from pre-trained full-precision models with ours for further verification.

## 2 RELATED WORK

Quantization is one of the most widely-used techniques for neural network compression (Courbariaux et al., 2015; Han et al., 2015; Zhu et al., 2016; Zhou et al., 2016; 2017; Mishra et al., 2017; Park et al.,

2017; Banner et al., 2018), with two types of training strategies: Post-Training Quantization directly quantizes a pre-trained full-precision model (He & Cheng, 2018; Nagel et al., 2019; Fang et al., 2020a;b; Garg et al., 2021); Quantization-Aware Training uses training data to optimize quantized models for better performance (Gysel et al., 2018; Esser et al., 2019; Hubara et al., 2020; Tailor et al., 2020). In this work, we focus on the latter one, which is explored in several directions. One area uses uniform-precision quantization where the model shares the same precision (Zhou et al., 2018; Wang et al., 2018; Choukroun et al., 2019; Gong et al., 2019; Langroudi et al., 2019; Jin et al., 2020a; Bhalgat et al., 2020; Chen et al., 2020; Yang et al., 2020; Darvish Rouhani et al., 2020; Oh et al., 2021). Another direction studies mixed-precision that determines bit-width for each layer through search algorithms, aiming at better accuracy-efficiency trade-off (Dong et al., 2019; Wang et al., 2019; Habi et al., 2020; Fu et al., 2020; 2021; Yang & Jin, 2020; Zhao et al., 2021a;b; Ma et al., 2021b). There is also binarization network, which only applies 1-bit (Rastegari et al., 2016; Hubara et al., 2016; Cai et al., 2017; Bulat et al., 2020; Guo et al., 2021). Despite the fact that quantization helps reduce energy consumption and inference latency, it is usually accompanied by performance degradation. To alleviate this problem, several methods are proposed.

One type of effort focuses on simulated quantization. The strategy is to leave some operations, e.g., BN, in full-precision for the stabilized training of quantized models (Choi et al., 2018; Esser et al., 2019; Jin et al., 2020b). Nevertheless, these methods limit the application of the quantized models on resource-demanding hardware, such as DSP, where full-precision arithmetic is not supported for accelerated computing (QCOM, 2019; Ho, 2015). To completely eliminate floating-point operations from the quantized model, integer-only quantization techniques emulate the full-precision multiplication by 32-bit integer multiplication followed by bit shifting (Jacob et al., 2018; Zhu et al., 2020; Wu et al., 2020; Yao et al., 2021; Kim et al., 2021). However, the calculation of INT32 multiplication in these works requires one more operation, which results in extra energy and higher latency (Gholami et al., 2021). In parallel, recent work (Jain et al., 2019) proposes to restrict all scaling factors as power-of-2 values for all weights and activations, which belongs to fixed-point quantization methods (Lin et al., 2016; Jain et al., 2019; Kim & Kim, 2021; Mitschke et al., 2019; Enderich et al., 2019b; Chen et al., 2017; Enderich et al., 2019a; Zhang et al., 2020; Goyal et al., 2021). This enables the model to only incorporate INT8 or even INT4 multiplications, followed by INT32 bit shifting. However, there still a lack of a thorough study of the benefits of using fixed-point arithmetic. Also, the power-of-2 scaling factors are directly determined from the training data without theoretical analysis and guidance. In this work, we give an extensive analysis, especially on the potential and theoretical principle of using fixed-point values for quantized models, and demonstrate that with proper analysis and design, a model quantized with only INT8 multiplication involved is able to achieve comparable and even better performance to the integer-only methods implemented with INT32 multiplication.

## 3   Analysis of Fixed-Point Representation

In this section, we first introduce the fixed-point multiplication (Smith et al., 1997; Tan & Jiang, 2018) and analyze the distribution of weight from different layers in a well-trained full-precision model (Sec. 3.1). We then investigate the statistical property of fixed-point numbers, and demonstrate the potential of approximating full-precision values by 8-bit fixed-point numbers with different formats (Sec. 3.2). After that, we study the relationship between standard deviation of random variables and the optimal fixed-point format with the smallest quantization error. Finally, we derive an approximated formula relating the standard deviation and fixed-point format, which is verified empirically and employed in our final algorithms (Sec. 3.3).

### 3.1   Advantages of Fixed-Point Arithmetic

Fixed-point number is characterized by its format, which includes both the word length indicating the whole bit-width of the number and the fractional length (FL) characterizing the range and resolution of the represented values (Smith et al., 1997). Fixed-point arithmetic—especially fixed-point multiplication—is widely utilized for applications in, e.g., digital signal processing (Smith et al., 1997; Tan & Jiang, 2018). Compared with integer or floating-point multiplication, fixed-point multiplication has two major characteristics: First, multiplying two fixed-point numbers is more efficient than multiplying two floating-point numbers, especially on resource-constrained devices such as DSP. Second, it is more powerful than its integer counterpart due to its versatility and the

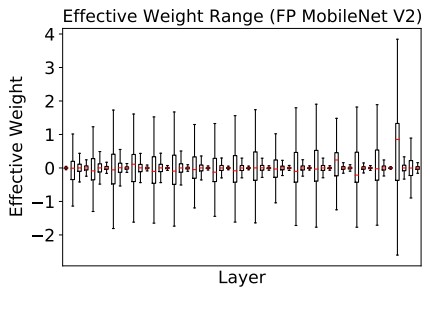
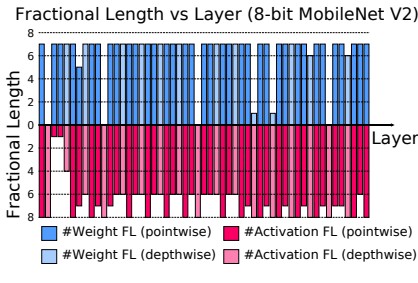

(a) Weight range.

(b) Weight and activation fractional length.

Figure 2: (a) Value range of effective weight (see Sec. 4.2) for a pre-trained full-precision (FP) model, and (b) fractional lengths of each layer for a well-trained fixed-point model for MobileNet V2.

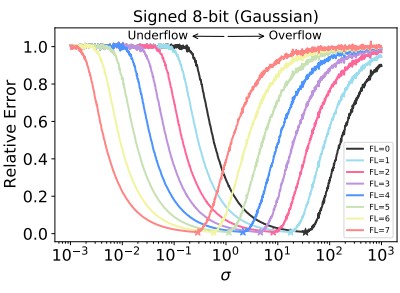
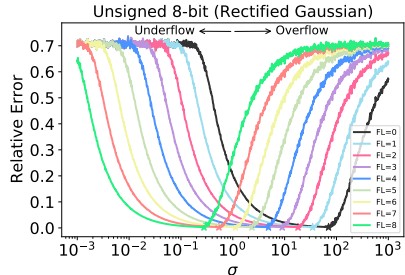

(a) Signed quant. for Gaussian R.V.

(b) Unsigned quant. for Rectified Gaussian R.V.

Figure 3: Representing potential for 8-bit signed (a) and unsigned (b) fixed-point numbers with different formats. The figures plot the relationship between relative quantization error and the standard deviation for different fixed-point formats. Both are experimented on zero-mean Gaussian random variables (R.V.), with ReLU applied on (b).

representative ability of fixed-point numbers (there can be tens of different implementations for fixed-point multiplication but only one for integer and floating-point ones (Smith et al., 1997)). This efficiency and versatility make fixed-point quantization a more appealing solution than integer-only quantization. Specifically, as shown in Fig. 2a, the scales of weights from different layers in a pre-trained full-precision model can vary in orders, ranging from less than $0.1$ to nearly $4$. Direct quantization with only integers inevitably introduces considerable quantization error, unless more precision and more operations are involved, such as using INT32 multiplication together with bit shifting for scaling as shown in Fig. 1c. On the other hand, employing fixed-point numbers has the potential to reduce quantization error without relying on high-precision multiplication, as weights and activations from different layers have the extra degree of using different formats during quantization. Indeed, as shown in Fig. 2b for a well-trained MobileNet V2 with 8-bit fixed-point numbers, the fractional lengths for weights and activations vary from layer to layer. This raises the question of how to determine the formats for each layer. In the following, we study this for 8-bit fixed-point models.

## 3.2 STATISTICAL ANALYSIS FOR FIXED-POINT FORMAT

For a predefined bit-width, integer, which is a special case of fixed-point numbers with zero fractional length, has a predefined set of values that it can take, which severely constrains the potential of integer-only quantization. On the other hand, fixed-point numbers, with an extra degree of freedom, i.e., the fractional length, are able to represent a much wider range of full-precision values by selecting the proper format, and thus they are more suitable for quantization. As an example, Fig. 3 shows the relative quantization error with 8-bit fixed-point values using different formats for a set of random variables, which are sampled from normal distributions (both signed and unsigned, with the latter processed by ReLU before quantization) with zero-mean and different standard deviations $\sigma$ (more experimental details in Appx. 7.2). From the experiments, we make the following two observations.

**Observation 1**: *Fixed-point numbers with different formats have different optimal representing regions, and the minimum relative error and optimal standard deviation (annotated as a star) varies*

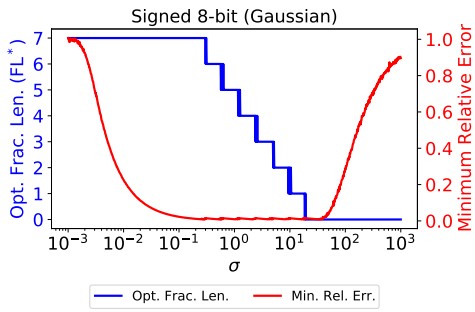

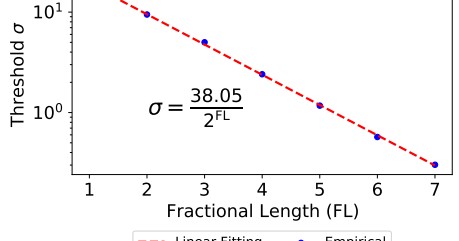

(a) Optimal fractional length and minimum relative error for signed quant.

(b) Relationship between threshold standard deviation and fractional length for signed quant.

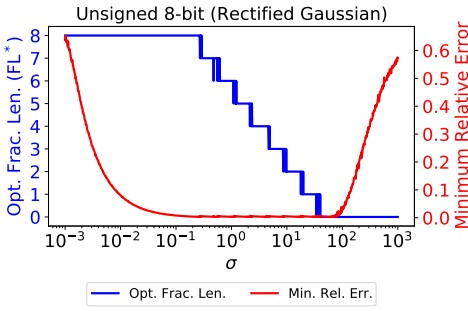

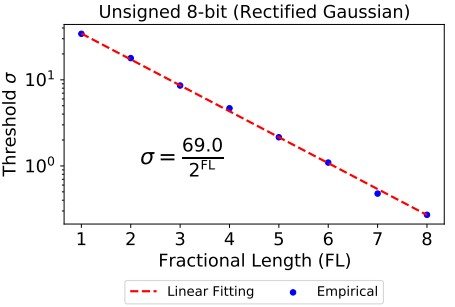

(c) Optimal fractional length and minimum relative error for unsigned quant.

(d) Relationship between threshold standard deviation and fractional length for unsigned quant.

Figure 4: Determining optimal fractional length from standard deviation. (a) and (c) illustrate optimal fractional length and minimum relative quantization error against standard deviation for signed and unsigned 8-bit fixed-point quantization for Gaussian and rectified Gaussian random variables. (b) and (d) show the relationship between threshold standard deviation and fractional length.

*for different fractional lengths (Fig. 3).* This is because the format controls the value magnitude and the representation resolution (the least significant bit).

**Observation 2:** *Larger fractional lengths are more robust to represent smaller numbers, while smaller fractional lengths are more suitable for larger ones.* For a given standard deviation, using small fractional length has the risk of underflow, while large fractional length might cause overflow issue. Specifically, integers (black curves in Fig. 4) are much more prone to underflow issues and have large relative errors for small enough values to quantize.

### 3.3 CHOOSING OPTIMAL FIXED-POINT FORMAT

With the above observations, we are interested in answering two questions:

*(1) Can we achieve a small fixed-point quantization error for a wide range of full-precision values by always using the optimal fractional length corresponding to the smallest relative error?*

To answer this, we first plot the smallest possible relative error amongst all the candidate fixed-point formats against the standard deviation. As shown in red lines from Fig. 4a and Fig. 4c, for zero-mean normal distribution, by always choosing the optimal fixed-point format, we are able to achieve a relative quantization error smaller than 1% for standard deviation with a range of order of at least around 3. For example, for signed quantization, the standard deviation can range from 0.1 to around 40 to achieve less than 1% error, and for unsigned quantization, the standard deviation can range from 0.1 to 100. The experiments verify our presumption that using fixed-point values with the optimal formats is able to achieve negligible quantization error.

*(2) Can we have a simple way to determine the optimal fractional length?*

To answer this, we plot the optimal fractional length from the statistics of the full-precision values against the standard deviation, as shown in the blue lines in Fig. 4a and Fig. 4c. We find that the

threshold $\sigma$ value corresponding to the jumping point is almost equidistant on the log scale of the standard deviation. This is expected as the representing region of different formats are differed by a factor of 2's exponents. Plotting the threshold standard deviation (on a log-scale) against the corresponding optimal fractional length (Fig. 4b and Fig. 4d), we find their relationship is almost linear, leading to the following semi-empirical approximating formulas to determine the optimal fractional length $\mathrm{FL}^*$ from the standard deviation (more discussion in Appendix 7.7):

$$\text{Signed}: \quad \mathrm{FL}^* \approx \lfloor \log_2 \frac{40}{\sigma} \rfloor, \qquad \text{Unsigned}: \quad \mathrm{FL}^* \approx \lfloor \log_2 \frac{70}{\sigma} \rfloor. \tag{1}$$

In the following, unless specifically stated, we use (1) to determine the fractional length for both weight and activation quantization. Note that we only calculate the standard deviation during training.

## 4 METHODS

In this section, we discuss our proposed training technique for neural network quantization with fixed-point numbers, where the formats of weights and activations in each layer are determined based on (1) during training. We first analyze how to unify PACT and fixed-point quantization (Sec. 4.1). Then we show how to quantize weights and activations, especially updating for BN running statistics and fractional lengths (Sec. 4.2). Finally, we discuss the necessity of relating scaling factors from two adjacent layers to calculate the effective weights for quantization, especially for residual blocks where some layers have several layers following them (Sec. 4.3).

### 4.1 UNIFYING PACT AND FIXED-POINT QUANTIZATION

To quantize a positive value $x$ with unsigned fixed-point number of format $(\mathrm{WL}, \mathrm{FL})$, where WL and FL denotes word length and fractional length for the fixed-point number, respectively, we have the quantization function fix_quant as:

$$\mathrm{fix\_quant}(x) = \frac{1}{2^{\mathrm{FL}}} \mathrm{round} \left( \mathrm{clip} \left( x \cdot 2^{\mathrm{FL}}, 0, 2^{\mathrm{WL}} - 1 \right) \right), \tag{2}$$

where clip is the clipping function, and $0 \leq \mathrm{FL} \leq \mathrm{WL}$ for unsigned fixed-point numbers. Note that fixed-point quantization has two limitations: overflow, which is caused by clipping into its representing region, and underflow, which is introduced by the rounding function. Both of these introduce approximation errors. To minimize the error, we determine the optimal fractional length for each layer based on the analysis in Sec. 3.3.

To achieve a better way to quantize a model using fixed-point numbers, we take a look at one of the most successful quantization techniques, PACT (Choi et al., 2018), which clips on the full-precision value with a learned clipping-level $\alpha$ before quantization:

$$\mathrm{PACT}(x) = \frac{\alpha}{M} \mathrm{round} \left( \frac{M}{\alpha} \mathrm{clip} \left( x, 0, \alpha \right) \right), \tag{3}$$

where $M$ is a pre-defined scale factor mapping the value from $[0, 1]$ to $[0, M]$. The formal similarity between (2) and (3) inspires us to relate them with each other as (more details in the Appx. 7.3):

$$\mathrm{PACT}(x) = \frac{2^{\mathrm{FL}} \alpha}{2^{\mathrm{WL}} - 1} \mathrm{fix\_quant}(\frac{2^{\mathrm{WL}} - 1}{2^{\mathrm{FL}} \alpha} x), \tag{4}$$

where we have set $M = 2^{\mathrm{WL}} - 1$, which is the typical setting. With this relationship, we can implement PACT and train the clipping-level $\alpha$ implicitly with fixed-point quantization.

### 4.2 UPDATING BN AND FRACTIONAL LENGTH

**Double Forward for BN Fusion**. To quantize the whole model with only 8-bit fixed-point multiplication involved, we need to tackle the scaling factor from BN layer, including both the weight and running variance. Specifically, we need to quantize the effective weight that fuses the weight of convolution layers with the weight and running variance from BN (Jacob et al., 2018; Yao et al., 2021). This raises the question of how to determine the running statistics during training. To solve this problem, we apply forward computation twice. *For the first forward*, we apply the convolution

using quantized input yet full-precision weight of the convolution layer, and use the output to update the running statistics of BN. In this way, the effective weight to quantize is available. Note there is no backpropagation for this step. *For the second forward*, we quantize the combined effective weight to get the final output of the two layers of convolution and BN and do the backpropagation.

**Updating Fractional Length**. Different from existing work that directly trains the fractional length (Jain et al., 2019), we define the fractional length for weight on-the-fly during training by inferring from current value of weight, using (1). For the fractional length of activation, we use a buffer to store and update the value with a momentum of $0.1$, similar to how to update BN running statistics. Once the fractional lengths are determined after training, we keep them fixed for inference.

### 4.3 RELATING SCALING FACTORS BETWEEN ADJACENT LAYERS

As shown in (4), there are still two extra factors during the quantization operation, which we denote as a fix scaling factor $\eta_{\text{fix}}$:

$$\eta_{\text{fix}} = \frac{2^{\text{FL}}\alpha}{2^{\text{WL}} - 1}. \tag{5}$$

Now $\alpha$ is a trainable parameter with full-precision, which means the fix scaling factor is also in full-precision. To eliminate undesired extra computation, we absorb it into the above-mentioned effective weights for quantization (Sec. 4.2). However, the fix scaling factor occurs twice, one for rescaling after quantization ($\eta_{\text{fix}}$) and the other for scaling before quantization ($1/\eta_{\text{fix}}$). To completely absorb it, we need to relate two adjacent layers. In fact, for a mapping that includes convolution, BN, and ReLU (more details are shown in Appx. 7.5), we apply PACT quantization to relate the activation between two adjacent layers as:

$$q_i^{(l+1)} = \text{fix\_quant}\left( \underbrace{\sum_{j=1}^{n^{(l)}} \frac{\gamma_i^{(l)}}{\sigma_i^{(l)}} \frac{\eta_{\text{fix}}^{(l)}}{\eta_{\text{fix}}^{(l+1)}} W_{ij}^{(l)} q_j^{(l)}}_{\text{Effective Weight}} + \underbrace{\frac{1}{\eta_{\text{fix}}^{(l+1)}}\left( \beta_i^{(l)} - \frac{\gamma_i^{(l)}}{\sigma_i^{(l)}} \mu_i^{(l)} \right)}_{\text{Effective Bias}} \right), \tag{6}$$

where $q$ is the fixed-point activation, $W$ the full-precision weight of the convolution layer, $i$ and $j$ the spatial indices, $n$ the total number of multiplication, and the superscript $(l)$ indicates the $l$-th block consisting of convolution and BN. $\gamma$, $\beta$, $\sigma$, $\mu$ are the learned weight, bias, running standard deviation, and running mean for the BN layer, respectively. Also, we set WL $= 8$ for all layers. As can be seen from (6), to obtain the final effective weight for fixed-point quantization, for the $l$-th Conv-BN block, we need to access the fix scaling factor, or equivalently, the clipping-level $\alpha$ and the activation fractional length FL, from its following $(l+1)$-th block(s). To achieve this, we apply two techniques.

**Pre-estimating Fractional Length**. As mentioned above, we determine the activation fractional length from its standard deviation. Also, (5) indicates that the fix scaling factor relies on such fractional length for each layer. However, in (6), we need the fix scaling factor from the next layer to determine the effective weight under quantization, which we have not yet updated. Thus, when calculating the effective weights during training, we use the activation fractional length stored in the buffer, instead of the one for quantizing the input of the next layer.

**Clipping-Level Sharing**. As shown in Fig. 5, for residual blocks, some layers have two following layers (which we also name as child layer). Since we need the fix scaling factor from the child layer to calculate the effective weight for the parent (see (6)), inconsistent fix scaling factors between all children layers will be a problem. To this end, we define one layer as master and force all its siblings to share its clipping-level. In fact, the best way is to share both the clipping-level and the fractional length among siblings, but we find sharing fractional length leads to considerable performance drop, especially for deep models such as MobileNet V2 and ResNet50. This is because the fractional lengths play two roles here: one is for the fix scaling factor, and the other is for the representing region (or equivalently the clipping-level). Using different fractional lengths effectively enables different clipping-levels (although only differ by a factor of power-of-2, see Appx. 7.6), which can be beneficial because the activation scales might vary from layer to layer. Moreover, breaking the constraint of sharing activation fractional length does not introduce much computational cost, as the value only differs in storing format, and typically the values are stored in 32-bit, i.e., the accumulation results are only quantized into 8-bit for multiplication. Note that when computing the effective weight

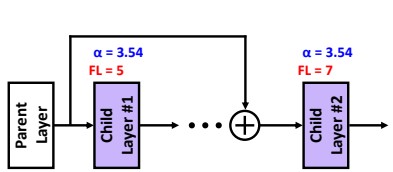

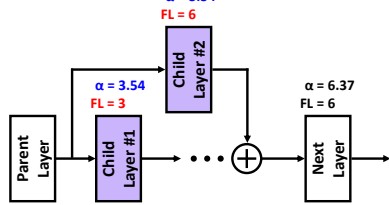

| (a) ResBlock with direct connection. | (b) ResBlock with downsampling. |

Figure 5: The illustration of residual connections. For a layer with several layers (named children layers) directly following it, we choose one to be master, and all its sibling layers use the master layer's clipping level. On the other hand, since using different fractional length only cause bit shifting or different fixed-point quantization formats, and the values are stored in 32-bit before quantized into 8-bit, we do not share the fractional formats to allow more degrees of freedom. The two figures show the case of direct residual connection (a) and that with downsampling convolution layer (b).

of the parent layer, we only use the master child's activation fractional length. For effective weight of each child layer and fixed-point quantization on its input, we use its own fractional length.

## 5 EXPERIMENTS

In this section, we present our results for various models on ImageNet (Deng et al., 2009) for classification task and compare the results with previous works that focus on quantization-aware training to verify the effectiveness of our method. We show the results for two sets of training. First, we discuss the conventional training method following Jin et al. (2020b). Second, we unify our method with one recent fine-tuning method that quantizes full-precision models with high accuracy (Yao et al., 2021). More detailed experimental settings are described in Appx. 7.1.

**Conventional training**. We first apply our method using conventional training (Choi et al., 2018; Esser et al., 2019; Jin et al., 2020b; Fu et al., 2021), where the quantized model is trained with the simplest setting as those for full-precision model (more details in Appx. 7.1). To verify the effectiveness of our method, we perform experiments on several models including ResNet18 and MobileNet V1/V2. As shown in Table 1, our method achieves the state-of-the-art results for all models. Additionally, we obtain comparable or even better performance than the full-precision counterparts.

Compared with previous works on simulated quantization (Choi et al., 2018; Park et al., 2018; Esser et al., 2019; Jin et al., 2020b; Fu et al., 2021) that requires full-precision rescaling after INT8 convolution, our approach is not only more efficient but also achieves better performance. On the other hand, compared with previous fixed-point quantization (Jain et al., 2019), our approach gives better results. This might partially due to that our method is based on a more systematic analysis, as explained above in Section 3.3.

Table 1: 8-bit quantization with conventional training for ResNet18 and MobileNet V1/V2b. Following Yao et al. (2021), we abbreviate Integer-Only Quantization as "Int", INT8-Multiplication-Only Quantization as "8-bit", the Baseline Accuracy as "BL", and Top-1 Accuracy as "Top-1". All models are for 8-bit weight and activation quantization. For MobileNet V2, we are using MobileNet V2b version as it is the most typical one.

(a) ResNet18

| Method | Int | 8-bit | BL | Top-1 |
| --- | --- | --- | --- | --- |
| Baseline (FP) | ✗ | ✗ | 70.3 | 70.3 |
| RVQuant (Park et al., 2018) | ✗ | ✗ | 69.9 | 70.0 |
| PACT (Choi et al., 2018) | ✗ | ✗ | 70.2 | 69.8 |
| LSQ (Esser et al., 2019) | ✗ | ✗ | 70.5 | 71.1 |
| CPT (Fu et al., 2021) | ✗ | ✗ | - | 69.6 |
| F8Net (ours) | ✓ | ✓ | 70.3 | **71.1** |

(b) MobileNet V1

| Method | Int | 8-bit | BL | Top-1 |
| --- | --- | --- | --- | --- |
| Baseline (FP) | ✗ | ✗ | 72.4 | 72.4 |
| PACT (Choi et al., 2018) | ✗ | ✗ | 72.1 | 71.3 |
| TQT (Jain et al., 2019) | ✓ | ✓ | 71.1 | 71.1 |
| SAT (Jin et al., 2020b) | ✗ | ✗ | 71.7 | 72.6 |
| F8Net (ours) | ✓ | ✓ | 72.4 | **72.8** |

(c) MobileNet V2b

| Method | Int | 8-bit | BL | Top-1 |
| --- | --- | --- | --- | --- |
| Baseline (FP) | ✗ | ✗ | 72.7 | 72.7 |
| PACT (Choi et al., 2018) | ✗ | ✗ | 72.1 | 71.7 |
| TQT (Jain et al., 2019) | ✓ | ✓ | 71.7 | 71.8 |
| SAT (Jin et al., 2020b) | ✗ | ✗ | 71.8 | 72.5 |
| F8Net (ours) | ✓ | ✓ | 72.7 | **72.6** |

To further understand the significance of our method, we plot the fractional lengths for weight and activation for each layer. Illustrated in Fig. 2b for MobileNet V2, we find that the fractional lengths for both weight and activation vary from layer to layer. Specifically, for weight quantization, since

some layers have relatively large value range of effective weight, especially some depthwise layers, small fractional length is necessary to avoid overflow issue. On the other hand, for layers with small weight scale, large fractional length has more advantages to overcome the underflow problem. The same conclusion also applies for the fractional length for activation. Indeed, for some early layers in front of depthwise convolution layer, the activation fractional length needs to be small, yet for the later-stages, larger fractional length is desired. This further verifies our finding that using different fractional lengths for layers with the same parent is critical for good performance, because layers at different depths might be siblings and requires different fractional lengths (see Fig. 5).

**Tiny fine-tuning on full-precision model**. Recent work (Yao et al., 2021) focus on investigating the potential of neural network quantization. To this end, they suggest to tiny fine-tune on a well-pretrained full-precision model with high accuracy. In this way, it might help to avoid misleading conclusion coming from improper comparison between weak full-precision models with strong quantized model. To further investigate the power of our method and compare it with these advanced techniques, we also apply our method and fine-tune on several full-precision models with high accuracy. Also, given the number of total fine-tuing steps is very small, we apply grid search to determine the optimal fractional lengths for this experiment. The results are listed in Table 2, and we can find that our method is able to achieve better performance than previous method (Yao et al., 2021), without time- and energy-consuming high-precision multiplication (namely dyadic scaling shown in Fig. 1c).

Our method reveals that the high-precision rescaling, no matter implemented in full-precision, or approximated or quantized with INT32 multiplication followed by bit-shifting (a.k.a. dyadic multiplication), is indeed un-

Table 2: 8-bit quantization with tiny fine-tuning on well-trained full-precision model. Following Yao et al. (2021), we abbreviate Integer-Only Quantization as "Int", INT8-Multiplication-Only Quantization as "8-bit", Layer-Wise Quantization as "Layer", the baseline accuracy as "BL", Top-1 Accuracy as "Top-1", and Top-1 Accuracy Drop with respect to the baseline as "Drop". We use two baselines for ResNet50, one from PytorchCV (Sémery, 2021) (Baseline #1) and another from Nvidia (Nvidia, 2021) (Baseline #2), and we use ResNet50b version. Note that the OMPQ (Ma et al., 2021b) is mixed-precision quantization.

(a) ResNet18

| Method | Int | 8-bit | Layer | BL | Top-1 | Drop |
|---|---|---|---|---|---|---|
| Baseline (FP) | ✗ | ✗ | - | 71.5 | 71.5 | - |
| HAWQ-V3 (Yao et al., 2021) | ✓ | ✗ | ✗ | 71.5 | 71.6 | 0.1 |
| HAWQ-V3 (Yao et al., 2021) | ✓ | ✗ | ✓ | 71.5 | 70.9 | -0.6 |
| OMPQ (Ma et al., 2021b) | ✓ | ✗ | ✗ | 73.1 | 72.3 | -0.8 |
| F8Net (ours) | ✓ | ✓ | ✓ | 73.1 | **72.4** | -0.7 |

(b) ResNet50b

| Method | Int | 8-bit | Layer | BL | Top-1 | Drop |
|---|---|---|---|---|---|---|
| Baseline #1 (FP) | ✗ | ✗ | - | 77.6 | 77.6 | - |
| HAWQ-V3 (Yao et al., 2021) | ✓ | ✗ | ✗ | 77.6 | 77.5 | -0.1 |
| HAWQ-V3 (Yao et al., 2021) | ✓ | ✗ | ✓ | 77.6 | 77.1 | -0.5 |
| F8Net (ours) | ✓ | ✓ | ✓ | 77.6 | **77.6** | 0.0 |
| Baseline #2 (FP) | ✗ | ✗ | - | 78.5 | 78.5 | - |
| HAWQ-V3 (Yao et al., 2021) | ✓ | ✗ | ✗ | 78.5 | 78.1 | -0.4 |
| HAWQ-V3 (Yao et al., 2021) | ✓ | ✗ | ✓ | 78.5 | 76.7 | -1.8 |
| F8Net (ours) | ✓ | ✓ | ✓ | 78.5 | **78.1** | -0.4 |

necessary and is not the key part for quantized model to have good performance. This is not well-understood in previous literature. Specifically, we demonstrate that by properly choosing the formats for weight and activation in each layer, we are able to achieve comparable and even better performance with 8-bit fixed-point numbers, which can be implemented more efficiently on specific hardwares such as DSP that only supports integer operation.

## 6  CONCLUSION

Previous works on neural network quantization typically rely on 32-bit multiplication, either in full-precision or with INT32 multiplication followed by bit-shifting (termed dyadic multiplication). This raises the question of whether high-precision multiplication is critical to guarantee high-performance for quantized models, or whether it is possible to eliminate it to save cost. In this work, we study the opportunities and challenges of quantizing neual networks with 8-bit only fixed-point multiplication, via thorough statistical analysis and novel algorithm design. We validate our method on ResNet18/50 and MobileNet V1/V2 on ImageNet classification. With our method, we achieve the state-of-the-art performance without 32-bit multiplication, and the quantized model is able to achieve comparable or even better performance than their full-precision counterparts. Our method demonstrates that high-precision multiplication, implemented with either floating-point or dyadic scaling, is not necessary for model quantization to achieve good performance. One future direction is to perform an in-depth statistical analysis of fixed-point numbers with smaller word-lengths for neural network quantization.

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

# 7 APPENDIX

## 7.1 MORE EXPERIMENTAL DETAILS

**More Details for Conventional Training**. For conventional training method, we train the quantized model initialized with a pre-trained full-precision one. The training of full-precision and quantized models shares the same hyperparameters, including learning rate and its scheduler, weight decay, number of epochs, optimizer, and batch size. For ResNet18 and MobileNet V1, we use an initial learning rate of $0.05$, and for MobileNet V2, it is $0.1$. We find the value of learning rate, i.e., $0.1$ and $0.05$, does not have much impact on the final performance. Totally, $150$ epochs of training are conducted, with cosine learning rate scheduler without restart. The warmup strategy is adopted with linear increasing ($\mathrm{batchsize}/256 \times 0.05$) (Goyal et al., 2017) during the first five epochs before cosine learning rate scheduler. The input image is randomly cropped to $224 \times 224$ and randomly flipped horizontally, and is kept as 8-bit unsigned fixed-point numbers with $\mathrm{FL} = 8$ and without standardization. For ResNet18 and MobileNet V1/V2, we use batch size of $2048$ and run the experiments on 8 A100 GPUs. The parameters are updated with SGD optimizer and Nesterov momentum with a momentum weight of $0.9$ without damping. The original structure of MobileNet V2 uses ReLU6 as its activation. Since our unified PACT and the fixed-point quantization already has clipping operation, and can be equivalently formulated with ReLU6 by rescaling weight or activation, we eliminate ReLU6 in our implementation.

**Discussion for Weight Decay**. We set weight decay to $4 \times 10^{-5}$, and find the weight decay scheme is critical for good performance, especially for the quantized model. We analyze weight decay for different models as follows:

- For ResNet18, we apply weight decay on all layers, including convolution, fully-connected, and BN layers.

- For MobileNet V1, previous methods only apply weight decay on conventional convolution and fully-connected layers, but not on depthwise convolution and BN (Howard et al., 2017). We find this leads to the overfitting problem, making some early convolution layers have large weights, which is not friendly for quantization. We further observe that some channels of some depthwise convolution layers have all zero inputs, due to some channels of previous layer become all negative and ReLU is applied afterwards, making the running statistics of the corresponding channels in the following BN layer almost zero. This breaks the regularization effect of BN (Luo et al., 2018). Since each output channel only depends on one input channel for depthwise convolution layers, the weights connecting them become uncontrolled, and the effective weights become large, leading to an overfitting problem. Applying weight decay on the depthwise convolution and BN layers helps to alleviate this problem, and the resulting effective weights become small.

- For MobileNet V2, we find overfitting plays the role of reducing the validation error (although the training error is lower), and applying weight decay on depthwise convolution or BN weights impairs the training procedure. The underlying reason might be related to the residual connecting structure of this model (note MobileNet V1 does not use residual connection).

In summary, we apply weight decay on all layers, including depthwise convolution and BN layers for ResNet18 and MobileNet V1, and do not apply weight decay on depthwise convolution and BN layers for MobileNet V2.

**More Details for Tiny Fine-tuning**. For tiny fine-tuning on full-precision models, we follow the same strategy proposed in Yao et al. (2021). Specifically, we use a constant learning rate of $10^{-4}$, with $500$ iterations of fine-tuning (or equivalently data ratio of around $0.05$ with batch size of $128$). Different from (Yao et al., 2021), we find fixed BN is not helpful, and we allow it to update during the whole fine-tuning step. As mentioned in Sec. 5, we apply grid search to determine the fractional lengths for both weight and input, as the training cost is very small and applying grid search does not introduce too much effort or training time. Also, since the original full-precision model uses the normalized input, we also apply normalization on the images and quantize images with signed fixed-point numbers (and format determined with grid search) before being fed into the first convolution layer of the model.

### 7.2 MORE DETAILS FOR STATISTICAL ANALYSIS

For the toy example in Fig. 3, we sample $10,000$ zero-mean Gaussian random variables with different standard deviations, and apply ReLU activation for the rectified Gaussian variables with unsigned quantization. The variables are then quantized with fixed-point quantization given in (2) and (9), respectively. We calculate the relative quantization error and plot against the standard deviation for each fixed-point format. Note that zero-mean is a reasonable simplifying assumption if we assume to neglect the impact of bias in BN for analysis purposes.

### 7.3 DERIVATION FOR FIXED-POINT AND PACT RELATION

Here we derive the relationship between PACT and fixed-point quantization shown in (4). Specifically, the PACT quantization in (3) can be formulated as follows for positive $\alpha$:

$$\text{PACT}(x) = \frac{\alpha}{M}\text{round}\left(\frac{M}{\alpha}\text{clip}\left(x, 0, \alpha\right)\right) \tag{7a}$$

$$= \frac{\alpha}{M}\text{round}\left(M\text{clip}\left(\frac{x}{\alpha}, 0, 1\right)\right) \tag{7b}$$

$$= \frac{\alpha}{M}\text{round}\left(\frac{M}{2^{\text{WL}}-1}\text{clip}\left(\frac{2^{\text{WL}}-1}{\alpha}x, 0, 2^{\text{WL}}-1\right)\right) \tag{7c}$$

$$= \frac{2^{\text{WL}}-1}{M}\frac{2^{\text{FL}}\alpha}{2^{\text{WL}}-1}\frac{1}{2^{\text{FL}}}\text{round}\left(\frac{M}{2^{\text{WL}}-1}\text{clip}\left(\frac{2^{\text{WL}}-1}{2^{\text{FL}}\alpha}x*2^{\text{FL}}, 0, 2^{\text{WL}}-1\right)\right). \tag{7d}$$

For $M = 2^{\text{WL}}-1$, which is the typical setting for quantization, we have:

$$\text{PACT}(x) = \frac{2^{\text{FL}}\alpha}{2^{\text{WL}}-1}\frac{1}{2^{\text{FL}}}\text{round}\left(\text{clip}\left(\frac{2^{\text{WL}}-1}{2^{\text{FL}}\alpha}x*2^{\text{FL}}, 0, 2^{\text{WL}}-1\right)\right). \tag{8}$$

Comparing with the expression for fixed-point quantization (2), we can immediately get (4).

### 7.4 DOUBLE SIDE QUANTIZATION FOR WEIGHT AND MOBILENET V2

In (2), we only give the formula for fixed-point quantization of unsigned case. For weight and activation from some layer without following ReLU nonlinearity (such as some layers in MobileNet V2), signed quantization is necessary, and the expression is similarly given as:

$$\text{fix\_quant}(x) = \frac{1}{2^{\text{FL}}}\text{round}\left(\text{clip}\left(x\cdot 2^{\text{FL}}, -2^{\text{WL}-1}+1, 2^{\text{WL}-1}-1\right)\right), \tag{9}$$

where $\text{clip}$ is the clipping function, and $0 \leq \text{FL} \leq \text{WL} - 1$.

### 7.5 DERIVATION OF EFFECTIVE WEIGHT

Here we derive the equation of effective weights relating two adjacent layers in Sec. 4.3. Specifically, for a Conv-BN-ReLU block with conventional PACT quantization using input clipping, quantization and dequantization, the general procedure can be described as

$$\text{Nonlinear:} \quad \widetilde{x}_i^{(l)} = \text{clip}\left(x_i^{(l-1)}, 0, \alpha^{(l)}\right), \tag{10a}$$

$$\text{Input Quant (uint8):} \quad \widehat{q}_i^{(l)} = \text{round}\left(\frac{M}{\alpha^{(l)}}\widetilde{x}_i^{(l)}\right), \tag{10b}$$

$$\text{Input Dequant:} \quad \widetilde{q}_i^{(l)} = \frac{\alpha^{(l)}}{M}\widehat{q}_i^{(l)}, \tag{10c}$$

$$\text{Conv:} \quad y_i^{(l)} = \sum_{j=1}^{n^{(l)}} W_{ij}^{(l)}\widetilde{q}_j^{(l)}, \tag{10d}$$

$$\text{BN:} \quad x_i^{(l)} = \gamma_i^{(l)}\frac{y_i^{(l)} - \mu_i^{(l)}}{\sigma_i^{(l)}} + \beta_i^{(l)} \tag{10e}$$

$$= \frac{\gamma_i^{(l)}}{\sigma_i^{(l)}} y_i^{(l)} + \left( \beta_i^{(l)} - \frac{\gamma_i^{(l)}}{\sigma_i^{(l)}} \mu_i^{(l)} \right), \tag{10f}$$

where $x$ is the input before clipping, $\widehat{q}$ is the integer input after quantization, $\widetilde{q}$ is the full-precision input after dequantization, clip is the clipping function, $\alpha$ is the clipping-level, $M = 2^{\mathrm{WL}} - 1$ is the scaling for quantization, $W_{ij}$ is weight from convolution layer, and $\gamma, \beta, \sigma, \mu$ are weight, bias, running standard deviation, and running mean from BN layer, respectively, and $i$ and $j$ are spatial indices. We first note that (10a), (10b) and (10c) can be combined as:

$$\widetilde{q}_i^{(l)} = \mathrm{PACT}(x_i^{(l-1)}) \tag{11a}$$

$$= \eta_{\mathrm{fix}}^{(l)} \mathrm{fix\_quant} \left( \frac{1}{\eta_{\mathrm{fix}}^{(l)}} x_i^{(l-1)} \right) \tag{11b}$$

$$= \eta_{\mathrm{fix}}^{(l)} q_i^{(l)}, \tag{11c}$$

where $q$ is the fixed-point activation and we have used the relationship given by (4) and the definition in (5). From this we can derive that:

$$q_i^{(l+1)} = \mathrm{fix\_quant} \left( \frac{1}{\eta_{\mathrm{fix}}^{(l+1)}} x_i^{(l)} \right) \tag{12a}$$

$$= \mathrm{fix\_quant} \left( \frac{1}{\eta_{\mathrm{fix}}^{(l+1)}} \left( \frac{\gamma_i^{(l)}}{\sigma_i^{(l)}} y_i^{(l)} + \left( \beta_i^{(l)} - \frac{\gamma_i^{(l)}}{\sigma_i^{(l)}} \mu_i^{(l)} \right) \right) \right) \tag{12b}$$

$$= \mathrm{fix\_quant} \left( \frac{1}{\eta_{\mathrm{fix}}^{(l+1)}} \left( \frac{\gamma_i^{(l)}}{\sigma_i^{(l)}} \sum_{j=1}^{n^{(l)}} W_{ij}^{(l)} \widetilde{q}_j^{(l)} + \left( \beta_i^{(l)} - \frac{\gamma_i^{(l)}}{\sigma_i^{(l)}} \mu_i^{(l)} \right) \right) \right) \tag{12c}$$

$$= \mathrm{fix\_quant} \left( \frac{1}{\eta_{\mathrm{fix}}^{(l+1)}} \left( \frac{\gamma_i^{(l)}}{\sigma_i^{(l)}} \sum_{j=1}^{n^{(l)}} W_{ij}^{(l)} \eta_{\mathrm{fix}}^{(l)} q_j^{(l)} + \left( \beta_i^{(l)} - \frac{\gamma_i^{(l)}}{\sigma_i^{(l)}} \mu_i^{(l)} \right) \right) \right) \tag{12d}$$

$$= \mathrm{fix\_quant} \left( \sum_{j=1}^{n^{(l)}} \frac{\gamma_i^{(l)}}{\sigma_i^{(l)}} \frac{\eta_{\mathrm{fix}}^{(l)}}{\eta_{\mathrm{fix}}^{(l+1)}} W_{ij}^{(l)} q_j^{(l)} + \frac{1}{\eta_{\mathrm{fix}}^{(l+1)}} \left( \beta_i^{(l)} - \frac{\gamma_i^{(l)}}{\sigma_i^{(l)}} \mu_i^{(l)} \right) \right), \tag{12e}$$

which is just (6).

## 7.6 Private Fractional Lengths Enabling Different Clipping-Levels

Here we analyze the effect of using private fractional lengths between sibling layers to indicate that this effectively enables private clipping-levels for them. In fact, the original PACT quantization step is given as

$$\widetilde{q} = \mathrm{PACT}(x) \tag{13a}$$

$$= \frac{2^{\mathrm{FL}} \alpha}{2^{\mathrm{WL}} - 1} \frac{1}{2^{\mathrm{FL}}} \mathrm{round} \left( \mathrm{clip} \left( \frac{2^{\mathrm{WL}} - 1}{2^{\mathrm{FL}} \alpha} x * 2^{\mathrm{FL}}, 0, 2^{\mathrm{WL}} - 1 \right) \right), \tag{13b}$$

where we have omitted layer and spatial indices for simplification. Now if we use private fractional lengths for sibling layers while require them to share the same clipping level, and use the master child's fractional length for calculating the effective weight in (6), denoting the fractional length of the master layer as $\mathrm{FL}^{\mathrm{m}}$, the above function becomes

$$\widetilde{q} = \frac{2^{\mathrm{FL}} \alpha}{2^{\mathrm{WL}} - 1} \frac{1}{2^{\mathrm{FL}}} \mathrm{round} \left( \mathrm{clip} \left( \frac{2^{\mathrm{WL}} - 1}{2^{\mathrm{FL}^{\mathrm{m}}} \alpha} x * 2^{\mathrm{FL}}, 0, 2^{\mathrm{WL}} - 1 \right) \right) \tag{14a}$$

$$= 2^{\mathrm{FL} - \mathrm{FL}^{\mathrm{m}}} \frac{2^{\mathrm{FL}} \alpha'}{2^{\mathrm{WL}} - 1} \frac{1}{2^{\mathrm{FL}}} \mathrm{round} \left( \mathrm{clip} \left( \frac{2^{\mathrm{WL}} - 1}{2^{\mathrm{FL}} \alpha'} x * 2^{\mathrm{FL}}, 0, 2^{\mathrm{WL}} - 1 \right) \right), \tag{14b}$$

where $\alpha' = 2^{\mathrm{FL}^{\mathrm{m}} - \mathrm{FL}} \alpha$. From this we see that using private fractional lengths effectively enables different clipping-levels between sibling layers, and the cost is only some bit shifting.

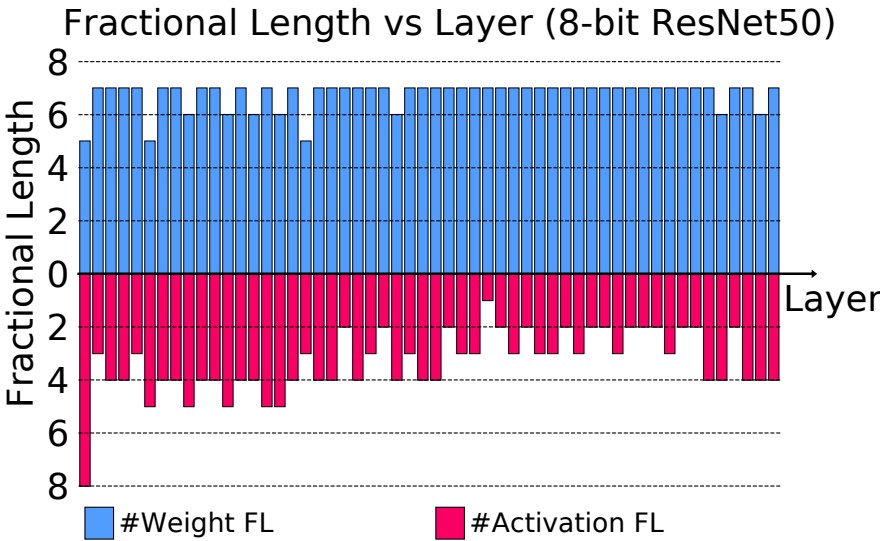

Figure 6: Fractional lengths of each layer for a well-trained fixed-point model for ResNet50.

Table 3: Analysis of the impact of the searching space for fractional length (ResNet50 on ImageNet).

| Method | Frac. Len. Range | BL | Top-1 |
|---|---|---|---|
| Baseline (FP) | - | 77.6 | 77.6 |
| F8Net (ours) | $6, 7, 8$ | 77.6 | 72.4 |
| F8Net (ours) | $0 - 8$ | 77.6 | **77.6** |

## 7.7 MORE DISCUSSION OF THE OPTIMAL FRACTIONAL LENGTH

Here we give some further discussion of using standard deviation to determine the optimal fractional length. The main reason is that standard deviation is a more robust statistics than others, such as dynamic range, and is an easily-estimated parameter for Gaussian distributed weights and pre-activations. Considering depth-wise convolution layers that contain much fewer weights and inputs, using robust statistics becomes essential as these layers might include weights or inputs with strange behavior, *e.g.*, the pre-activation values of some channels become all negative with large magnitude. Therefore, the standard deviation is more suitable and robust than the dynamic range.

## 7.8 FRACTIONAL LENGTH FOR RESNET50

Here we provide more results of fractional lengths distribution in Fig. 6 for the well-trained ResNet50 with 8-bit fixed-point numbers finetuned from the Baseline #2 in Table 2b. As we can see, the optimal fractional lengths are layer-dependent and their distribution is highly different from those in MobileNet V2 (as shown in Fig. 2b). Specifically, for MobileNet V2, some layers have vanishing weight fractional lengths and less than $4\%$ of all layers have an activation fractional length less than 4, while for ResNet50, more than $88\%$ of all layers have an activation fractional length that is less or equal to 4.

## 7.9 ANALYZING SEARCHING SPACE OF FRACTIONAL LENGTHS

In the main paper, we adopt the largest possible searching space for the fractional lengths of 8-bit fixed-point. As shown in Fig. 2b, many layers have a fractional length less than 4, either for input or

weight. Here we study whether it is possible to use only fractional lengths between 6 and 8. To this end, we finetune on ResNet50b using the Baseline #1. The results are listed in Table 3, from which we find that restricting the fractional lengths between 6 to 8 significantly impacts the performance of the final quantized model, as the top-1 accuracy drops from 77.6% to 72.4%.

