# OpenReview forum: "F8Net: Fixed-Point 8-bit Only Multiplication for Network Quantization"
_ICLR.cc/2022/Conference — ICLR 2022 Oral_

### Official Review · Reviewer_jedd · 2021-11-01

**Correctness:** 4
**Technical Novelty And Significance:** 3
**Empirical Novelty And Significance:** Not applicable
**Recommendation:** 6
**Confidence:** 4

**Main Review:**

Strengths:
 - The paper is well written and easy to understand. Technical details are clear.
 - The idea seems practical and straightforward to implement in real systems

Weaknesses:
 - The paper claims that the quantized training flow uses only 8-bit fixed-point multiplications. However, calculating the fractional length requires the standard deviation, which needs to be done in float? The two-pass method for batch norm also uses float in the first pass. It's not clear if the technique is suitable on a specialized accelerator that performs only fixed-point multiplication. Instead the technique seems to target quantization-aware training (QAT) on GPUs for producing a quantized model. If this is the case, the results seem weak - the improvement over SOTA is very small, and the models tested are all small scale. The results are not enough to convince me that this is a significant contribution to QAT literature.
 - The novelty of the formulas to compute the fractional length seems weak. It's common in QAT to scale the tensor by dividing by the max value or dividing by some clip value. The scaled tensor then has range [-1, 1] and maps to a fixed-point representation with no integer portion. It's not clear what is gained by using stddev to estimate the fractional length instead. I would like to see some small experiments specifically comparing these ways to handle the dynamic range of a float tensor. In my mind they should achieve similar accuracy results.
 - It's not clear to me what combining PACT with quantization achieves. PACT is already meant to be used with quantization, and it's not clear what we gain by combining the two instead of doing PACT, followed by quantization. Again, some small experiments comparing the two would be useful.


**Summary Of The Paper:**

The authors propose a new quantization flow to train DNNs using only 8-bit fixed-point multiplications. They show that 8-bit fixed point can represent different exponent ranges based on fractional length, thus choosing the right fractional length is critical. They then empirically derive a formula to calculate the optimal fractional length for a tensor based on its standard deviation. The authors combine PACT (learnable clip threshold) with fixed-point quantization, and propose a two-pass method to handle batch norm. They can achieve a small accuracy improvement (<1%) when training from scratch or fine-tuning on ImageNet using a variety of small models (ResNet-18 and MobileNet variants), compared to other quantization-aware training methods.


**Summary Of The Review:**

The authors propose a method to estimate the optimal fractional length of a fixed-point format given a float tensor, as well as other minor improvements to a quantization-aware training flow such as (1) combining PACT and quantization, (2) a two-pass method to handle batch norm. However, the novelty of the proposed methods is low, and it's not clear to me what improvements they offer over existing standards. The end-to-end accuracy improvement of the authors' technique is fairly small. As a result I rate the paper marginally below acceptance. If the authors can provide experiments that show specifically how each of their proposals improve over existing standard, then I would consider raising my score.

EDIT: The authors have addressed most of my concerns and I will raise my score to a 6. I still think that the formulas to compute the fractional length are not novel. The idea of estimating the dynamic range of the tensor and using that to determine the quantization range is well known.

---

> ### Author Response · Authors · 2021-11-20
> **Author Responses to Reviewer jedd (Part 1/2)**
>
> **We would like to thank you for your thoughtful review and valuable feedback. We have modified the paper based on the reviewer's suggestion. In the following, we provide more details on how to use fixed-point quantization during inference (Q1), clarification of novelty (Q2), reasons for using different fixed-point formants (Q3), and the comparison between our work and PACT (Q4). We hope our response can help address the reviewer's concern.**
>
> ---
>
> **Q1: Clarification on standard deviation, two-pass method for batch norm, and accelerators for fixed-point.**
>
> - **The goal of this paper.** We work on quantization aware training, instead of quantized training. Our goal is to get a quantized model for accelerated inference after deployment. Reducing the training efforts is not the goal of our work.
>
> - We agree calculating standard deviation and batch normalization running statistics requires 32-bit floating-point operations. However, those steps are only **conducted during training**, and **none of them are required for inference**. When the training is done, all fractional lengths for both weights and inputs of all layers have already been determined. We can directly run the model with 8-bit fixed-point arithmetic without any necessity to calculate the standard deviations during inference. We have clarified this important point following Eqn. 1 in Section 3.3 in the revised paper.
>
> - Special accelerators designed specifically for fixed-point are not required to run our quantized model. Compared with other integer only quantization, we remove the 32-bit multiplication without changing other quantization pipelines (Figure 1 (c) vs. Figure 1 (d)). We can directly deploy our quantized networks on integer accelerators. For example, we test MobileNet V1 and V2 on CPU with only integer operation involved, and we can get top-1 accuracy of 72.8% and 72.6%, which are the same as the one we get from GPU.
>
> ---
>
> **Q2: Clarification on novelty of our approach.**
>
> As mentioned by Reviewer 7TRc, our approach for 8-bit only multiplication with fixed-point numbers is creative, and the insights from our work will inform research on network quantization for quite some time. As shown in the final experiments in the paper, our theory can help to guide the practice, unify fixed-point quantization with PACT, and lead to state-of-the-art performance.
>
> ---
>
> **Q3: Scaling tensor into [-1, 1] and mapping with fixed-point representation without integer portion, the benefits gained by using standard deviation to estimate the fractional lengths, and small experiments comparing these two approaches.**
>
> We appreciate the reviewer’s discussion on scaling and directly using fixed-point numbers without the integer part. The method proposed by the reviewer is innovative but might have some problems as follows.
>
> - 32-bit scaling operations are still required to scale the values into [-1, 1], which are what we are trying to eliminate in our work. We are not sure how to merge them to avoid these calculations and only use fixed-point numbers without integer part (namely, without determining the format for the fixed-point numbers).
>
> - We are not sure if we understand clearly what the reviewer wants us to compare with and what small experiments to conduct, but we do the experiment as follows: scale (with 32-bit floating-point division) the weights and inputs with their maximum values to map them into [-1, 1] and [0, 1] respectively, and quantize the values directly with fractional lengths of 7 and 8, respectively (namely, we have something like +/-0.1010110 and 0.11011100), and we finetune the model. We find the ResNet50 does not converge correctly, and the top-1 accuracy is only 1%. We guess the main reason is that there are some specific requirements on the weights and activation ranges for the model to converge during training.

---

> > ### Comment · Reviewer_jedd · 2021-11-29
> > **Changing my score to a weak accept**
> >
> > The authors have addressed most of my concerns and I will raise my score to a 6. I still think that the formulas to compute the fractional length are not novel. The idea of estimating the dynamic range of the tensor and using that to determine the quantization range is well known.

---

> > > ### Author Response · Authors · 2021-11-29
> > > **Further Response to Reviewer jedd**
> > >
> > > Dear Reviewer jedd,
> > >
> > > Thank you for checking our responses and raising the score. We appreciate your time and efforts. It is our great pleasure to know our efforts help address your concerns.
> > >
> > > We agree that using the standard deviation (or dynamic range) to determine the fractional length (or quantization range) for fixed-point quantization might be discussed in other literature (maybe the studies in digital communication or digital signal processing). However, based on our knowledge, we are the first to incorporate the standard deviation for choosing the fixed-point format into the training of quantized neural networks. Our method can avoid more complicated learning-based methods and achieve better results (as also mentioned by Reviewer 7TRc).
> > >
> > > Thanks again for your time.
> > >
> > > Best,
> > >
> > > Authors

---

> ### Author Response · Authors · 2021-11-20
> **Author Responses to Reviewer jedd (Part 2/2)**
>
> **Q4: PACT followed by quantization; the benefits gained by combining PACT with fixed-point quantization; small experiments comparing these two approaches.**
>
> We agree that PACT is meant to be used for quantization and is a powerful quantization technique, especially for activations. This is the reason why we focus on PACT and try to formulate PACT with fixed-point quantization.
>
> We assume that the reviewer asked the *conventional quantization method that uses PACT for activation quantization, *i.e.*, clipping the values followed by quantization*. Suppose we use PACT followed by quantization, which is the conventional approach (*e.g.*, in [1]). In that case, it requires scaling before quantization and rescaling after quantization, which introduces extra computation with high precision (*e.g.*, 32-bit) and leads to more latency and energy consumption. This belongs to simulated quantization (Figure 1 (b)). Also, implementing these with floating-point values can not fully leverage the integer accelerators. On the other hand, using fixed-point, we only need bit-shifting, as illustrated in Figure 1 (d) in the paper, to save energy and latency.
>
> To extensively examine the advantages of fixed-point quantization over other quantization methods (including simulated and 32-bit multiplication-based integer-only quantization), we conduct some small experiments as suggested by the reviewer, and compare the two approaches from three different aspects, *i.e.*, latency efficiency, energy efficiency, and accuracy performance.
>
> - **Small experiments on latency comparison.** For small experiments, here we list the results of latency comparison on CPU between convolution with 8-bit integer multiplication and that additionally using 32-bit integer multiplication for scaling. We run some simple tests (batch size is 1) using iPhone XR CPU on some toy modules to get the hardware evidence of the efficiency of our method. We use XNNPACK [2], which is a highly optimized library. We find that 8-bit multiplication is able to accelerate the convolution by around 6~8%.
>
>   **Table A. Latency improvement on iPhone XR CPU with XNNPACK.**
> > | Input Channels | Output Channels | Kernel Size | Groups | Height | Width | Latency Improvement (%) |
> > |:--------------:|:---------------:|:-----------:|:------:|:------:|:-----:|:-----------------------:|
> > |        3       |        8        |      3      |    1   |   160  |  160  |           6.57          |
> > |        3       |        16       |      3      |    1   |   160  |  160  |          12.34          |
> > |       480      |       480       |      3      |   480  |   10   |   10  |           6.33          |
> > |       480      |       480       |      3      |   480  |   16   |   16  |           8.36          |
>
> - **Evidence for energy and area cost saving.** As demonstrated in Fig.7 in [3] (Page 12), latency and energy consumption for 8-bit integer multiplications are much less than the 32-bit floating-point or integer multiplications. For instance, the 32-bit integer multiplication consumes 15x more energy than 8-bit integer multiplication, and the area cost saving (or equivalently, roughly proportional to the number of gates required) is around 12.4x.
>
> - **Accuracy comparison.** In Tables 1 & 2, we give accuracy comparisons for PACT followed by quantization and demonstrate that our method can give better results. For example, on MobileNetV1, we get 72.8 while PACT gets 71.3, and on MobileNetV2, we get 72.7 while PACT gets 71.1.
>
> We hope our explanation can help make this clearer.
>
> ---
>
> **References**
>
> [1] Jungwook Choi, Zhuo Wang, Swagath Venkataramani, Pierce I-Jen Chuang, Vijayalakshmi Srinivasan, and Kailash Gopalakrishnan. PACT: Parameterized clipping activation for quantized neural networks. arXiv preprint arXiv:1805.06085, 2018.
>
> [2] https://github.com/google/XNNPACK
>
> [3] Amir Gholami, Sehoon Kim, Zhen Dong, Zhewei Yao, Michael W Mahoney, and Kurt Keutzer. A survey of quantization methods for efficient neural network inference. arXiv preprint arXiv:2103.13630, 2021.

---

> ### Author Response · Authors · 2021-11-28
> **Author Response to Reviewer jedd**
>
> Dear Reviewer jedd,
>
> We appreciate your reviewing efforts to improve our work.
>
> We follow your initial comments to provide additional results, specifically how each of our proposals improve over existing standard. For example, we conducted the suggested small experiment using the scaling method proposed by the reviewer for quantization, which could not converge even though we spent several days working on it. We also provide more hardware evidence, including latency and energy efficiency, to help clarify the motivation of our 8-bit fixed-point quantization. We hope our response can help further demonstrate that our approach is crucial for achieving further accelerated quantized neural networks with high performance by eliminating the unnecessary 32-bit multiplication widely-adopted in conventional integer-only quantization techniques.
>
> As the discussion deadline is approaching, we would sincerely appreciate it if you could kindly let us know whether our response is satisfactory and has addressed your concerns. It will be our great pleasure if you would consider updating your review or score.
>
> Best,
>
> Authors

---

### Official Review · Reviewer_1v2J · 2021-11-02

**Correctness:** 3
**Technical Novelty And Significance:** 3
**Empirical Novelty And Significance:** 2
**Recommendation:** 5
**Confidence:** 5

**Main Review:**

strengths:
+ The authors consider a practical issue from the perspective of DNN hardware implementation. The problem is well-defined.
+ The paper is easy to understand. The solution is presented clearly.

Weakness:

- I am not convinced that 8-bit fix-point format is cheaper to implement than the other numeric formats. Quantization has been studied very extensively by the ML community, and numerous numeric formats have been applied to facilitate the implementation of the DNN. (e.g., binary quantized DNN, Logarithm quantization [a1], block floating point [a2], etc). The authors should better motivate this fix-point format by providing some hardware evidence. Without this, it is hard to convince the reviewer about the usefulness of this work.
- Equation 1 is based on the empirical approximation with Gaussian distribution, not the real DNN trace. Given the importance of this equation, authors should better justify the correctness of this empirical approximation with real DNN trace.
- This work assumes the ReLU is used for nonlinear operation. How to handle other types of nonlinear operations (e.g., leaky ReLU)?
- In the paper, the author mentioned that they use previously stored activation fractional length for pre-estimating the fractional length for the following layer. What does the previously stored activation mean? Is it the activation from the last training batches?
- Observation 1 and 2 is obvious. It is nice to have some real empirical results to demonstrate this,  but I think they should be described very briefly.

[a1] Oh, Sangyun, et al. "Automated Log-Scale Quantization for Low-Cost Deep Neural Networks." Proceedings of the IEEE/CVF Conference on Computer Vision and Pattern Recognition. 2021.

[a2] Darvish Rouhani, Bita, et al. "Pushing the limits of narrow precision inferencing at cloud scale with microsoft floating point." Advances in Neural Information Processing Systems 33 (2020).

**Summary Of The Paper:**

This paper describes a novel quantization framework that involving only fix-point 8-bit multiplication for DNN execution. The paper first highlights the advantages of the fixed-point numeric format. The paper then conducts some statistical study and derive an empirical formula to relate the fraction length of the fix-point representation with the standard deviation of the value distribution. After that, the paper introduces a novel approach to determine the right format for each layer during the forward propagation of the training. The proposed solution, F8Net, has been evaluation on ImageNet using multiple DNN structures (e.g., MobileNet, ResNet).


**Summary Of The Review:**

Overall, I think this paper lacks some motivation for the fix-point format. Given the fact that quantization has been studied extensively by the previous work, I am not convinced that 8-bit fix-point can achieve a better hardware efficiency than the previous work. Some hardware results may be helpful.

---

> ### Author Response · Authors · 2021-11-20
> **Author Responses to Reviewer 1v2J (Part 1/2)**
>
> **We appreciate the thoughtful review and comments from the reviewer. We have modified the paper based on the reviewer's suggestion, clarified the meaning of previously stored activation fractional lengths (Q5), and simplified the description of Observations 1 & 2 (Q6). Other concerns are also addressed in the following. We hope our efforts can help further reveal the potential and significance of 8-bit multiplication only fixed-point quantization.**
>
> ---
>
> **Q1: 8-bit fixed point vs other numeric formats.**
>
> We agree that binarization and other numeric formats such as logarithm quantization [1] and block floating point [2] are also able to accelerate neural networks. However, to the authors’ best knowledge, current work of binarization that achieves comparable performance than the full-precision counterparts only binarizes on weights and activations in convolutional and fully-connected layers, but not all layers implemented with binary operation [3-5]. For example, typically they require 32-bit floating-point scaling or batch normalization layers, which are not friendly to hardware platforms that only support integer operations. Also, such a combination of binary operation and floating-point operations (scaling) are less efficient, mainly because the accumulation results are stored with 32-bit integers and thus an extra type conversion is necessary. For network compression, our work is orthogonal to these branches of research and can be combined with these methods. However, we would like to appreciate the suggestion of the reviewer and have included both works [1-2] in our revision as references.
>
> ---
>
> **Q2: Hardware evidence that motivates 8-bit fixed-point format.**
>
> Better efficiency and less energy consumption is the main motivation for our proposal to eliminate the inefficient 32-bit multiplication in deep networks and use only 8-bit multiplication.
>
> - **Evidence for better latency efficiency.** We run some simple tests using iPhone XR CPU on some toy modules to get the hardware evidence of the efficiency of our method. We use XNNPACK [6], which is a highly optimized library. We compare our method with previous integer only quantization as shown in the following (batch size is 1).
>
>   **Table A. Latency improvement on iPhone XR CPU with XNNPACK.**
> > | Input Channels | Output Channels | Kernel Size | Groups | Height | Width | Latency Improvement (%) |
> > |:--------------:|:---------------:|:-----------:|:------:|:------:|:-----:|:-----------------------:|
> > |        3       |        8        |      3      |    1   |   160  |  160  |           6.57          |
> > |        3       |        16       |      3      |    1   |   160  |  160  |          12.34          |
> > |       480      |       480       |      3      |   480  |   10   |   10  |           6.33          |
> > |       480      |       480       |      3      |   480  |   16   |   16  |           8.36          |
>
>   It can be seen that our method is able to achieve acceleration for several different settings, even under such a simple testing environment, which is not optimal for our implementation using fixed-point arithmetic.  Actually, to fully employ the advantage of our technique, it might be better to develop a thoroughly new library, which is not easy to finish immediately. As discussed in [7], DSP engineering with fixed-point arithmetic involves a large amount of tradeoff between complexity, efficiency and accuracy, making fixed-point values more promising than other implementations. An industry-level library to fully leverage the efficiency of our fixed-point quantization is beyond the scope of this paper.
>
> - **Evidence for energy and area cost saving.** As demonstrated in Fig.7 in [8] (Page 12), latency and energy consumption for 8-bit integer multiplications are much less than 32-bit floating-point or integer multiplications. For instance, 32-bit integer multiplication consumes 15x more energy than 8-bit integer multiplication, and the area cost saving (or equivalently, roughly proportional to the number of gates required) is around 12.4x.
>
> ---
>
> **Q3: Justify the correctness of Equation 1 with real DNN trace.**
>
> We want to emphasize that **we indeed use real DNN traces** in all our experiments. Actually, the motivation of using Gaussian for theoretical analysis to guide practice is that typical DNN models have weights distributed as Gaussian, due to the Central Limit Theorem. Also, Eqn. (1) is a theoretical analysis to guide practice, and our latter experiments demonstrate that such theoretical analysis leads to good results. We humbly think that using this theoretical analysis is an advantage of our method.

---

> ### Author Response · Authors · 2021-11-20
> **Author Responses to Reviewer 1v2J (Part 2/2)**
>
> **Q4: Handling of other types of nonlinear operations.**
>
> We agree that more complicated nonlinear activations such as PReLU or GELU might require more attention to deal with instead of naive application of our method directly. Actually, based on our knowledge, PReLU is less frequently adopted in quantized networks. It has been used in binary models, which as described above, is a different research topic of our work. Nevertheless, if we indeed need to use it, it might be a better practice to separate that layer as an individual layer (although not as efficient as ReLU) and use 8-bit fixed-point numbers to approximate the parameter in such layers. For other more complicated functions, approximating them with polynomial function in 8-bit fixed-point arithmetic might be a good option, following the method proposed in [9], or we might be able to directly quantize the input of these activation functions with 8-bit fixed-point values, and use look-up-table to calculate it more efficiently, as we only need to store 255 different values for each of this (and totally $8\times255$ values for all different formats for all layers).
>
> ---
>
> **Q5: Meaning of previously stored activation fractional lengths.**
>
> We appreciate the reviewer for pointing out this point. As mentioned in the paper on the last sentence of Section 3, we determine the fractional length of input for each layer from its standard deviation. Also, when we calculate the effective weight in Eqn. 6, we need the fix scaling factor from the next layer that relies on its input fractional length via Eqn. 5, which has not been updated. Thus, for such calculation during training, we use the input fractional lengths stored in the buffer, instead of the one used when quantizing the input of the next layer. We have made minor modifications in the revised paper.
>
> ---
>
> **Q6: Observations 1 and 2 should be described very briefly.**
>
> We appreciate the suggestion from the reviewer and agree that the description can be abbreviated. We include them mainly to motivate our method and to make the analysis clear to readers who are not familiar with fixed-point quantization. We rephrase the two observations in the revised paper as suggested by the reviewer.
>
> ---
>
> **References**
>
> [1] Oh, Sangyun, et al. "Automated Log-Scale Quantization for Low-Cost Deep Neural Networks." Proceedings of the IEEE/CVF Conference on Computer Vision and Pattern Recognition. 2021.
>
> [2] Darvish Rouhani, Bita, et al. "Pushing the limits of narrow precision inferencing at cloud scale with microsoft floating point." Advances in Neural Information Processing Systems 33 (2020).
>
> [3] Rastegari et al., “XNOR-Net: ImageNet Classification using Binary Convolutional Neural Networks.” European Conference on Computer Vision. 2016.
>
> [4] Liu et al., “ReActNet: Towards Precise Binary Neural Network with Generalized Activation Functions.” European Conference on Computer Vision. 2020.
>
> [5] Zhang et al., “Dynamic Binary Neural Network by Learning Channel-Wise Thresholds”, arXiv preprint arXiv: 2110.05185, 2021.
>
> [6] https://github.com/google/XNNPACK
>
> [7] Steven W Smith et al. The scientist and engineer’s guide to digital signal processing. 1997.
>
> [8] Amir Gholami, Sehoon Kim, Zhen Dong, Zhewei Yao, Michael W Mahoney, and Kurt Keutzer. A survey of quantization methods for efficient neural network inference. arXiv preprint arXiv:2103.13630, 2021.
>
> [9] Sehoon Kim, Amir Gholami, Zhewei Yao, Michael W Mahoney, and Kurt Keutzer. I-bert: Integer only bert quantization. arXiv preprint arXiv:2101.01321, 2021.

---

> ### Author Response · Authors · 2021-11-28
> **Author Response to Reviewer 1v2J**
>
> Dear Reviewer 1v2J,
>
> Thank you again for your reviewing efforts to improve our work.
>
> We provide additional hardware evidence, including latency and energy efficiency, that helps clarify the motivation of our 8-bit fixed-point quantization. Your other questions are also answered in our response, and the suggestions are reflected in the revised paper. We hope our response can help further demonstrate that our approach is crucial for achieving further accelerated quantized neural networks with high performance by eliminating the unnecessary 32-bit multiplication widely-adopted in conventional integer-only quantization techniques.
>
> As the discussion deadline is approaching, we would sincerely appreciate it if you could kindly let us know whether our response is satisfactory and has addressed your concerns. It will be our great pleasure if you would consider updating your review or score.
>
> Best,
>
> Authors

---

> ### Comment · Reviewer_1v2J · 2021-11-29
> **Post rebuttal**
>
> Thanks for your detailed response. For Q1, I think there are already some papers to describe the fully-quantized DNN model (e.g., [a1], [a2]).
> I am not if F8Net can outperform these solutions. In addition, I do not think floating-point batchnorm and activation is that expensive to implement. For example, we could use table lookup to prestore the results of the nonliner functions in the buffer, so no computing hardware is required.
>
> Therefore, I decide to keep my original score (5).
>
> [a1] Peng, Peng, et al. "Fully integer-based quantization for mobile convolutional neural network inference." Neurocomputing 432 (2021): 194-205.
> [a2] Yao, Zhewei, et al. "Hawq-v3: Dyadic neural network quantization." International Conference on Machine Learning. PMLR, 2021.

---

> > ### Author Response · Authors · 2021-11-29
> > **Further Response to Reviewer 1v2J**
> >
> > Dear Reviewer 1v2J,
> >
> > Thank you for your feedback.
> >
> > First of all, **our models are fully quantized neural networks.**  There is no floating operation required in our models during inference time. We can directly deploy our quantized networks on integer accelerators. For example, we test MobileNet V1 and V2 on CPU with only integer operation involved, and we can get top-1 accuracy of 72.8% and 72.6%, which are the same as the one we get from GPU. *We humbly think that how to implement floating-point operations on integer-only hardware is not related to our work.* Also, the implementation of floating-point operations on integer-only hardware might not fully utilize the integer accelerators.
> >
> > Second, as mentioned by the reviewer, [a2] shows fully quantized models. We further reduce the computation from [a2] by eliminating the 32-bit multiplication (Fig. 1 (c) *vs.* Fig. 1 (d)). We conduct an extensive comparison with [a2] on ImageNet for ResNet 18 / 50 (shown in Tab. 2 of our paper) and demonstrate better results. For example, the performance of [a2] on ResNet 50 using layer-wise quantization is 76.7%, while ours is 78.1%.
> >
> > Third, we provide the table (Table A in our response) to show the better latency efficiency of our method (8-bit multiplication only fixed-point quantization) compared with [a2] on hardware.
> >
> > Fourth, our method achieves better classification accuracy than [a1] on ResNet 18. We will add the comparison and the reference paper to the revised version.
> >
> > We would like to kindly note that our models are fully quantized, and do not require 32-bit multiplication as used by other integer-only quantization methods (such as [a2]). We sincerely hope that the implementation and optimization of floating-point operations on integer-only hardware (which is not related to our work) will not affect the reviewer for the review and rating.
> >
> > Thanks again for your time.
> >
> > Best,
> >
> > Authors
> >
> > [a1] Peng, Peng, et al. "Fully integer-based quantization for mobile convolutional neural network inference." Neurocomputing 432 (2021): 194-205.
> >
> > [a2] Yao, Zhewei, et al. "Hawq-v3: Dyadic neural network quantization." International Conference on Machine Learning. PMLR, 2021.

---

> > > ### Author Response · Authors · 2021-11-30
> > > **Follow-up with Further Response to Reviewer 1v2J**
> > >
> > > Dear Reviewer 1v2J,
> > >
> > > We would like to thank you again for your suggestions and feedback to improve our work.
> > >
> > > We provide additional explanations to help clarify our work. Given the deadline for response is today, we would sincerely like to use this opportunity to see if our responses are sufficient and any concern remains.
> > >
> > > Thanks again for your time.
> > >
> > > Best,
> > >
> > > Authors

---

### Official Review · Reviewer_xQht · 2021-11-02

**Correctness:** 3
**Technical Novelty And Significance:** 2
**Empirical Novelty And Significance:** 2
**Recommendation:** 5
**Confidence:** 4

**Main Review:**

Strengths:
1- The proposed numerical format is evaluated in variant of benchmarks which mean the proposed approach can be deployed and generalized for different models
2- Combining the PACT and fixed point is interesting approach.

Weaknesses:
1- The correlation between standard division and fraction bit-width is interesting but it is obvious and in my opinion it is not required. The fraction length can be selected based on dynamic range and I think the reason behind the constant in the equation 1 is the correlation between standard division and dynamic range (DR=3ST). If we closely look at the figure 3, we find most of these standard divisions ( more than 1 ) is not useful for your case study, since dynamic range of most parameters is less than one for most of the layers. This small dynamic range of parameters explains why most of the layers parameters are using 6 to 8 bit fractions in figure 2. This brings a doubt on the motivation of why using the fixed point instead of INT8 approach?
2- The state of approach like dyadic quantization [1] and linear quantization with bit shift [2] performs DNN inference with 8-bit quantization without 32-bit INT multiplication. How is your approach different compared to these studies?
3- The author claims the performance of his approach is better than state of the art. However, the 32-bit baseline is different and the degradation accuracy should be reported in Tables 1 and 2. For instance, in ResNet18 result the degradation of accuracy in HAWQ-V3 is less than the proposed approach.
4- The new approach needs to compare with pervious work for 4-bit quantization. It is difficult to understand the advantages of this approach in compared to pervious work in 8-bit?
[1] Yao, Zhewei, et al. "Hawq-v3: Dyadic neural network quantization." International Conference on Machine Learning. PMLR, 2021.
[2] Langroudi, Hamed F., et al. "Cheetah: Mixed low-precision hardware & software co-design framework for dnns on the edge." arXiv preprint arXiv:1908.02386 (2019).

**Summary Of The Paper:**

The paper proposes a low-precision DNN inference models with 8-bit fixed point. To realize the number of fraction bits, the author uses the variance of DNN parameters and combines it with PACT approach in QAT. The new approach is evaluated in various neural networks such as MobileNet V1/V2 and ResNet18/50 on ImageNet for image classification and the result are mostly par with the state of the art approaches.

**Summary Of The Review:**

As a reviewer mentioned, the benefits and motivation of the proposed works is not obvious and the experimental result have not shown the advantages of this approach compared to state of the art works in 8-bit integer quantization. Therefore, I recommended this manuscript is marginally rejected.

---

> ### Author Response · Authors · 2021-11-20
> **Author Responses to Reviewer xQht (Part 1/2)**
>
> **Thanks for your valuable feedback and comments. We have reflected them in the revised paper to improve our work. In the following, we clarify more about the motivation of our approach including empirical analysis and explain differences from existing methods (Q1-Q4), provide additional experimental evidence as per reviewer’s feedback (Q5), and discuss the future direction of extending to INT4 case (Q6). We hope our response can further demonstrate the strengths of our method.**
>
> ---
>
> **Q1: Dynamic range vs. standard deviation for determining fractional length.**
>
> Thanks for proposing the relationship between dynamic range and standard deviation. We agree their relationship suggested by the reviewer could be a good explanation of why using standard deviation for determining the fractional length is an effective way. We can indeed try to use the dynamic range or other statistics (such as mean absolute error which we have also tried in our early exploration) to determine the fractional length.
>
> We finally decided to use standard deviation because it is a more **robust statistics** than dynamic range and is an easily-estimated parameter for Gaussian distributed weights and pre-activations. Considering depth-wise convolution layers that contain much fewer weights and inputs, using robust statistics becomes essential as these layers might include weights or inputs with strange behavior, *e.g.*, the pre-activation values of some channels become all negative with large magnitude. Therefore, the standard deviation is more suitable and robust than the dynamic range. We have included the explanation relating the standard deviation and dynamic range in the revised paper (Appendix 7.7) to give a more intuitive description.
>
> ---
>
> **Q2: Most of the standard deviations in Figure 3 are not useful and the small dynamic range of parameters explains why most of the layer parameters are using 6 to 8 bit fractions in Figure 2.**
>
> - The reason that we use 8 fractional lengths is that the 8-bit fixed-point value has 8 options. Using all the 8 options can enlarge the searching space and lead to better performance.
>
> - We would like to emphasize that the fractional lengths shown in Figure 2 **cannot** represent the fraction lengths used in other neural networks. Different fractional lengths besides 6 or 8 can be useful for other neural networks. For example, in Appendix 7.8 of the revised paper, we illustrate the fractional lengths with respect to the layers for ResNet50, and we can find that over 85% of layers have an input fractional length that is less or equal to 4.
>
> - To further verify that all the 8 fractional lengths can be useful, we run experiments by training models with fractional lengths **only from 6 to 8**. The comparison result for ResNet50 is listed below (also included in Appendix 7.9 of the revised paper). We can find that restricting the fractional length can significantly impair the accuracy of the final model.
>
>   **Table A. Analysis of the impact of the searching space for fractional length (ResNet50 on ImageNet).**
> > | Fractional Length Range | Top1 Acc (%) |
> > |:-----------------------:|:------------:|
> > |           {6, 7, 8}           |     72.4     |
> > |          {0, 1, 2, 3, 4, 5, 6, 7, 8}          |     **77.6**     |
>
> ---
>
> **Q3: Motivation of using fixed-point instead of INT8.**
>
> We are not sure if we understand the question or if the reviewer can kindly give more explanation on the point. Based on our understanding, the question is *why using fixed-point quantization instead of INT8 quantization*.
>
> To answer the above question, we first note that our approach is indeed **INT8 quantization for neural networks**. Other INT8 quantization methods still require an additional scaling operation implemented with INT32 multiplication and bit-shifting, as illustrated in Figure 1 (c) in the paper. Our approach belongs to INT8 quantization methods for neural networks and is more advanced than others in a way that **we do not need additional INT32 multiplication** (Figure 1 (d) in the paper).
>
> ---
>
> **Q4: Difference between our approach and previous work of dyadic quantization [1] and linear quantization with bit-shift [2].**
>
> - Ref. [1] employs 32-bit multiplication during dequantization (Page 3, Figure 1 in [1]), while our work does not require 32-bit multiplication (Figure 1 of our paper). Additionally, we unify PACT into fixed-point quantization without extra clipping operation.
>
> - Thank you for mentioning Ref. [2]. It studies the hardware-software co-design and uses three numeric types (namely floating, fixed, and posit numbers) to represent weights and activations, providing an extensive study on various models and datasets. The operation in the paper still requires some 32-bit multiplication, *e.g.*, the scaling in Eqn. (5). Therefore, the paper is orthogonal to our work. It is a good paper, and we have referenced it in the revision.

---

> > ### Comment · Reviewer_7TRc · 2021-11-30
> > **This is convining**
> >
> > I find the presented data and additional experiments convincing and, for me, they alleviate any concerns raised by reviewer xQht.

---

> ### Author Response · Authors · 2021-11-20
> **Author Responses to Reviewer xQht (Part 2/2)**
>
> **Q5: Performance comparison with different 32-bit baseline.**
>
> - We follow the practice of the most recent literature [1] to use models with the highest accuracy to perform quantization, since weak models could lead to misleading accuracy for the quantized models. Using strong baselines are specifically mentioned and suggested in [1] (Page 7, Section 4, the first paragraph, and the last two sentences).
>
> - We use the same baseline on ResNet50 as the one used in [1]. We achieve more improvement than [1], especially for performing layer-wise scaling (Ours 77.6 vs. [1]’s 77.1).
>
> - We also verify our results on ResNet18 using the same baseline model from HAWQ-V3 (top-1 accuracy of 71.5%) with layer-wise quantization, and we can get the same result (70.9%).
>
> - As suggested by the reviewer, we include the accuracy difference in the revised paper.
>
> ---
>
> **Q6: Extending our methods for 4-bit quantization.**
>
> Thank you for mentioning this. Extending our work to INT4 is a good direction, and we leave it as future work, mainly because INT4 is beyond the scope of our work as we focus on using INT8 multiplication only in this paper. We study INT8 multiplication only quantization mainly because INT8 is the most widely supported and optimized low-precision integer format for general purpose devices such as CPU and GPU. For INT4 quantization, we need some further extension of our method to support it, such as clipping suggested in [3]. Note that in [3] the authors also need to use INT8 for activation quantization. Using INT4 only to quantize both weights and activations still requires further in-depth study to achieve acceptable performance.
>
> ---
>
> **References**
>
> [1] Zhao et al., "Hawq-v3: Dyadic neural network quantization." International Conference on Machine Learning. PMLR, 2021.
>
> [2] Langroudi, Hamed F., et al. "Cheetah: Mixed low-precision hardware & software co-design framework for dnns on the edge." arXiv preprint arXiv:1908.02386, 2019.
>
> [3] Sambhav R Jain, Albert Gural, Michael Wu, and Chris H Dick. Trained quantization thresholds for accurate and efficient fixed-point inference of deep neural networks. arXiv preprint arXiv:1903.08066, 2019.

---

> ### Author Response · Authors · 2021-11-28
> **Author Response to Reviewer xQht**
>
> Dear Reviewer xQht,
>
> Thank you again for your thoughtful feedback to improve our work.
>
> We update the paper with additional results (Appendix 7.7, 7.8, 7.9 in the revision) to explain the reason for using fixed-point numbers for quantization, and that all 8-bit fraction lengths are useful. We hope our response can help further demonstrate that our approach is crucial for achieving further accelerated quantized neural networks with high performance by eliminating the unnecessary 32-bit multiplication widely-adopted in conventional integer-only quantization techniques.
>
> As the discussion deadline is approaching, we would sincerely appreciate it if you could kindly let us know whether our response is satisfactory and has addressed your concerns. It will be our great pleasure if you would consider updating your review or score.
>
> Best,
>
> Authors

---

> > ### Comment · Reviewer_xQht · 2021-11-28
> > **Post Rebuttal**
> >
> > I appreciate the authors' responses to my comments. After carefully reading your responses and other reviewers’ comments, I decide not to change the score. The response to Q4 is not satisfactory. As mentioned by other reviewers and I suggested, the author needs to compare (quantitively) their approach in terms of hardware metrics and accuracy with references [1,2,3,4,5,6] that used an 8-bit fixed point. The hardware evidence mentioned by the author is not convincing.
> >
> > [1] Zhao et al., "Hawq-v3: Dyadic neural network quantization." International Conference on Machine Learning. PMLR, 2021.
> > [2] Langroudi, Hamed F., et al. "Cheetah: Mixed low-precision hardware & software co-design framework for dnns on the edge." arXiv preprint arXiv:1908.02386, 2019
> > [3] Hashemi, Soheil, et al. "Understanding the impact of precision quantization on the accuracy and energy of neural networks." Design, Automation & Test in Europe Conference & Exhibition (DATE), 2017. IEEE, 2017.
> > [4] Gysel, Philipp, et al. "Ristretto: A framework for empirical study of resource-efficient inference in convolutional neural networks." IEEE transactions on neural networks and learning systems 29.11 (2018): 5784-5789.
> > [5] Judd, Patrick, et al. "Proteus: Exploiting numerical precision variability in deep neural networks." Proceedings of the 2016 International Conference on Supercomputing. 2016.
> > [6] Lin, Darryl, Sachin Talathi, and Sreekanth Annapureddy. "Fixed point quantization of deep convolutional networks." International conference on machine learning. PMLR, 2016.

---

> > > ### Author Response · Authors · 2021-11-29
> > > **Further Response to Reviewer xQht**
> > >
> > > Dear Reviewer xQht,
> > >
> > > We thank the reviewer for feedback.
> > >
> > > Generally, the focus of our work is **how to train the quantized neural network using fixed-point formats to achieve good performance** (*e.g.*, high accuracy on ImageNet), instead of focusing on providing a solution that can dramatically improve hardware efficiency. Most references provided by the reviewer are from the domain of Computer Architecture or Electronic Design Automation (EDA), which are other domains that mainly focus on how to improve hardware efficiency. Some of these references (*e.g.*, [5]) even do not provide any result on classification accuracy. We would like to kindly emphasize that these are **orthogonal research domains**, and it is not easy to provide a thorough and fair comparison between papers from different domains.
> > >
> > > More specifically, in the following, we would like to provide more facts, explanations, and comparisons with the provided references to help understand the advantages of our approach.
> > >
> > > - First, the network performance (*e.g.*, the classification accuracy) on [2-6] is not as good as the recent works. Also, [2,3,5,6] did not report the performance of large quantized models on ImageNet. Specifically, only results on small datasets are reported in [2, 3, 6] mostly with significant performance degradation, and [6] only reports error rate of the top-5 accuracy for a shallow network (with 5 conv layers and 2 fc layers). It is hard for us to implement all their methods on ImageNet.
> > >
> > > - Second, the work that has the best performance among all the references provided by the reviewer is [1]. We conduct an extensive comparison with [1] on ImageNet for ResNet 18 / 50 (shown in Tab.2 of our paper). For example, the performance of [1] on ResNet 50 using layer-wise quantization is 76.7%, while ours is 78.1%.
> > >
> > > - Third, we conduct the experiments on hardware to show the advantages of our method over the integer-only quantization method [1] for one convolution layer. We run some simple tests using iPhone XR CPU with XNNPACK [7], which is a widely-used library. As shown in the following table, our method can achieve faster inference speed than [1] for the convolution operation. We want to kindly note that an industry-standard library to fully leverage the efficiency of our fixed-point quantization is beyond the scope of this paper.
> > >
> > >   **Table A. Latency improvement on iPhone XR CPU with XNNPACK.**
> > > > | Input Channels | Output Channels | Kernel Size | Groups | Height | Width | Latency Improvement (%) |
> > > > |:--------------:|:---------------:|:-----------:|:------:|:------:|:-----:|:-----------------------:|
> > > > |        3       |        8        |      3      |    1   |   160  |  160  |           6.57          |
> > > > |        3       |        16       |      3      |    1   |   160  |  160  |          12.34          |
> > > > |       480      |       480       |      3      |   480  |   10   |   10  |           6.33          |
> > > > |       480      |       480       |      3      |   480  |   16   |   16  |           8.36          |
> > >
> > > - Fourth, as demonstrated in Fig.7 in [8] (Page 12), latency and energy consumption for 8-bit integer multiplications are much less than 32-bit floating-point or integer multiplications. For instance, 32-bit integer multiplication consumes $15\times$ more energy than 8-bit integer multiplication, and the area cost saving (or equivalently, roughly proportional to the number of gates required) is around $12.4\times$.
> > >
> > > Overall, we would again thank the reviewer for providing feedback.
> > >
> > > Best,
> > >
> > > Authors
> > >
> > > [1] Zhao et al., "Hawq-v3: Dyadic neural network quantization." ICML. PMLR, 2021.
> > >
> > > [2] Langroudi, Hamed F., et al. "Cheetah: Mixed low-precision hardware & software co-design framework for dnns on the edge." arXiv preprint arXiv:1908.02386, 2019.
> > >
> > > [3] Hashemi, Soheil, et al. "Understanding the impact of precision quantization on the accuracy and energy of neural networks." Design, Automation & Test in Europe Conference & Exhibition (DATE), 2017. IEEE, 2017.
> > >
> > > [4] Gysel, Philipp, et al. "Ristretto: A framework for empirical study of resource-efficient inference in convolutional neural networks." IEEE transactions on neural networks and learning systems 29.11 (2018): 5784-5789.
> > >
> > > [5] Judd, Patrick, et al. "Proteus: Exploiting numerical precision variability in deep neural networks." Proceedings of the 2016 International Conference on Supercomputing. 2016.
> > >
> > > [6] Lin, Darryl, Sachin Talathi, and Sreekanth Annapureddy. "Fixed point quantization of deep convolutional networks." ICML. PMLR, 2016.
> > >
> > > [7] https://github.com/google/XNNPACK
> > >
> > > [8] Amir Gholami, Sehoon Kim, Zhen Dong, Zhewei Yao, Michael W Mahoney, and Kurt Keutzer. A survey of quantization methods for efficient neural network inference. arXiv preprint arXiv:2103.13630, 2021.

---

> > > > ### Author Response · Authors · 2021-11-30
> > > > **Follow-up with Further Response to Reviewer xQht**
> > > >
> > > > Dear Reviewer  xQht,
> > > >
> > > > We would like to thank you again for your suggestions and feedback to improve our work.
> > > >
> > > > We provide additional results and comparisons to help alleviate your most recent concerns. Given the deadline for response is today, we would sincerely like to use this opportunity to see if our responses are sufficient and any concern remains.
> > > >
> > > > Thanks again for your time.
> > > >
> > > > Best,
> > > >
> > > > Authors

---

> > > > > ### Comment · Reviewer_xQht · 2021-11-30
> > > > > **Response to Authors**
> > > > >
> > > > > I appreciate the authors' responses to my comments. The paper that I mentioned are consider both hardware and accuracy. The result mentioned in Reference 5 is also mentioned accuracy ( The result mentioned in Table 2 is obtained within 1% degradation of accuracy of FP32). It is important to consider both accuracy and hardware metric when the paper highlighting removing INT32 as one of its contributions. The experiments on hardware that you mentioned are not satisfactory since you run two fixed-point quantization ( your method and [1] ) in the XNNPACk which is optimized for floating-point quantization. It is no doubt that the paper has a contribution but still, in my opinion, it needs to be completed by adding a meaningful comparison with previous works with considering both accuracy and hardware metrics. Therefore, I decide to keep my original score (5).

---

> > > > > > ### Author Response · Authors · 2021-11-30
> > > > > > **Clarification on XNNPACK and the Reference**
> > > > > >
> > > > > > Dear Reviewer xQht
> > > > > >
> > > > > > We really appreciate your timely response.
> > > > > >
> > > > > > Our understanding of your main concern is that we utilize XNNPACK [1] for latency comparison. We want to explain more reasons for choosing XNNPACK.
> > > > > >
> > > > > > First, we would like to kindly emphasize that **XNNPACK is an industry-level library that is highly optimized for integer quantization**. XNNPACK is built upon QNNPACK [2], and QNNPACK is a highly optimized library for low-precision, high-performance network inference on mobile.  XNNPACK can even achieve faster inference for integer quantized networks than QNNPACK. More details can be found in [3].
> > > > > >
> > > > > > Second, we humbly think the comparison experiments using XNNPACK are more convincing than simply running the comparison results using other libraries, such as PyTorch, because the latency improvement can be verified on an integer-quantization optimized library.
> > > > > >
> > > > > > Third, we would like to kindly mention that XNNPACK is not optimized for fixed-point quantization. Still, we can observe latency improvements of our approach over the integer quantization ([4]).
> > > > > >
> > > > > > Finally, as for [5], the reviewer mentioned *The result mentioned in Table 2 is obtained within 1% degradation of accuracy of FP32*. However, we do not find the accuracy in Tab. 2, as mentioned by the reviewer. We are wondering how to use Tab. 2 to get the accuracy for different models, or any other information in [5] that can obtain the accuracy on ImageNet directly.
> > > > > >
> > > > > > Thanks again for your time.
> > > > > >
> > > > > > Best,
> > > > > >
> > > > > > Authors
> > > > > >
> > > > > >
> > > > > > [1] https://github.com/google/XNNPACK
> > > > > >
> > > > > > [2] https://github.com/pytorch/QNNPACK
> > > > > >
> > > > > > [3] https://blog.tensorflow.org/2021/09/faster-quantized-inference-with-xnnpack.html
> > > > > >
> > > > > > [4] Zhao et al., "Hawq-v3: Dyadic neural network quantization." ICML. PMLR, 2021.
> > > > > >
> > > > > > [5] Judd, Patrick, et al. "Proteus: Exploiting numerical precision variability in deep neural networks." Proceedings of the 2016 International Conference on Supercomputing. 2016.

---

> > > > > > > ### Author Response · Authors · 2021-12-01
> > > > > > > **Follow-up with Clarification on XNNPACK**
> > > > > > >
> > > > > > > Dear Reviewer xQht,
> > > > > > >
> > > > > > > We hope our provided additional information about XNNPACK in the latest response could help answer why XNNPACK is a suitable library for testing the integer quantization models.
> > > > > > > *XNNPACK is designed for speeding up not only for floating-point models, but also integer quantized models.*
> > > > > > >
> > > > > > > We sincerely hope that our explanation could help alleviate the reviewer's major concern about using XNNPACK for latency analysis.
> > > > > > >
> > > > > > > Besides the latency, we also provide quantitative analysis for energy and area cost saving in our previous response. We hope that the provided hardware evidence, though not the focus of our paper, could help strengthen our work. We would really appreciate it if the reviewer would see the value of our work and the suggested experiments we run.
> > > > > > >
> > > > > > > Thanks again for your time and feedback.
> > > > > > >
> > > > > > > Best,
> > > > > > >
> > > > > > > Authors

---

> > > ### Comment · Reviewer_7TRc · 2021-11-30
> > > **I think your perspective is too hardware oriented.**
> > >
> > > While I agree partially with reviewer xQht concerns, I feel that the response misses the point of the paper. One can take a hardware perspective and a quantization error perspective.
> > >
> > > While the hardware perspective is very important to apply 8-bit training to computer vision networks such methods utterly fail for more complicated models like transformers. The second perspective, that this paper is taking, is that of relative quantization errors and how these can be minimized for fixed-point data types. This perspective is very important to make the first steps to approach the more difficult problems that lie ahead, such as 8-bit training of transformers.
> > >
> > > A counterpoint might be, that this paper should then either show 8-bit training results for transformers or a comprehensive analysis for all 8-bit data types (INT8). To that, I would say, that these problems are so complex that intermediate studies like presented here are necessary to gather the proper insights to be able to tackle such problems. It is unreasonable to ask for a more extensive comparison when the comparison presented in the paper is already extensive.
> > >
> > > I learned a lot from this paper and I will apply its insights directly to my research. While some insights are found also in some of the papers that you mentioned, I have never seen a such clearly presented perspective on relative quantization errors for fixed-point data types.

---

> > > > ### Comment · Reviewer_xQht · 2021-11-30
> > > > **Response to Reviewer 7TRc**
> > > >
> > > > Dear Reviewer 7TRc,
> > > >
> > > > Thanks for your comments. If we only consider the quantization error, we have better numerical format options than the fixed point. The benefit of fixed-point numerical format is more on the hardware perspective rather than quantization error.  Therefore considering both hardware metrics and accuracy are important.

---

> > > > > ### Author Response · Authors · 2021-11-30
> > > > > **Response to Reviewer xQht**
> > > > >
> > > > > Dear Reviewer xQht,
> > > > >
> > > > > As mentioned by the reviewer: *If we only consider the quantization error, we have better numerical format options than the fixed point. The benefit of fixed-point numerical format is more on the hardware perspective rather than quantization error*.  This is a well-known fact, and we are also aware of that. This is the reason why we focus on accuracy instead of hardware efficiency, as it is a more challenging problem for fixed-point quantization.
> > > > >
> > > > > Meanwhile, per the reviewer’s request, we also provide hardware evidence to show the advantages of our method, including latency efficiency and energy-saving.
> > > > >
> > > > > Thanks,
> > > > >
> > > > > Authors

---

### Official Review · Reviewer_7TRc · 2021-11-03

**Correctness:** 4
**Technical Novelty And Significance:** 4
**Empirical Novelty And Significance:** 4
**Recommendation:** 10
**Confidence:** 5

**Main Review:**

Pro:
This is a very strong paper and I have not seen any paper of such quality in this space for some time. The analysis of relative error and normal distributions gives a strong empirical and theoretical basis for the approach used in this work and provides deep insight into what the optimal fixed-point data type in each layer would look like. This analysis alone will inform future research in this area. Usually, such data types would be learned. It is very creative to abandon this approach and instead start from the analysis. Later this approach is merged with learned quantization through unification with PACT.

Overall, the results are outstanding and appear to be reliable given the extensive analysis of the approach.

Con:
Some details might not be accessible to readers that do not have an extensive background, especially pre-estimation of the fraction length and the sharing of the clipping thresholds for residual connections. It might be beneficial to trim the conclusion and add more context for these sections.


==============================UPDATE===================================

All the other reviewers take a very hardware-oriented perspective. I feel like they are not seeing the broader picture of this work which goes much beyond its initial applications. Many insights of this work are directly applicable for mixed-precision hardware where most, but not all operations are done in some 8-bit data type. From my own experience, I know that most 8-bit training methods fail when scaled up to models with billions of parameters.  For that area, quantization precision is the most important unsolved problem. This is a very important problem since transformer models keep getting larger and larger and accelerating training through 8-bit methods can make it more feasible to train these models. This work is one of the only ones in this space that makes some headway by its excellent analysis and impressive results. Many of the insights of the paperw are immediately applicable to my research.

Overall, I disagree with the perspective of the other reviewers and I still believe this is an outstanding paper. As such, I keep my score of 10.

**Summary Of The Paper:**

The paper analyzes the relationship between relative quantization errors and fixed-point formats for zero-centered normal distributions and finds a linear model which fits the best exponent length of the fixed-point data type given a standard deviation. These insights are then unified with parameterized clipping activation (PACT) to normalize incoming floating point data into the desired fixed point range. To handle the network with sole 8-bit multiplications a forward pass in floating precision is used to compute batch norm statistics for the main 8-bit forward/backward pass. Additional adjustments are made between successive layers and residual layers which rely on reusing some statistics of the previous layer.


**Summary Of The Review:**

The approach taken in this paper is very creative and the insights gained from this paper will inform research in this space for quite some time. As such, I strongly recommend acceptance of this work. I would be happy for this work to be selected as an oral presentation.

---

> ### Author Response · Authors · 2021-11-20
> **Author Responses to Reviewer 7TRc**
>
> Thank you for identifying the significance of our work and your strong recommendation for acceptance. We appreciate your acknowledgement that our approach for 8-bit only multiplication with fixed-point numbers is creative, our results are outstanding and reliable, and the insights from our work will inform research on network quantization for quite some time. In addition, our work has the potential application in innovative computing systems, such as computing-in-memory (CIM) systems where 32-bit integer multiplication is too bulky and inefficient that cannot be affordable, yet 8-bit integer multiplication only is a promising solution.
>
> We agree that the linear model found from the theoretical analysis of relative error and standard deviation distributions can empirically help decide the fixed-point format, instead of using learning-based approaches. Through building PACT and fixed-point quantization under one forward pass, unnecessary clip operation used in PACT can be abandoned to make the whole quantization formula much neater.
>
> As suggested, we make some revisions on the paper to give more details on pre-estimating fractional length and clipping level sharing, and simplify the conclusion. We hope our revision can make these points clearer and would like to appreciate your constructive suggestions to improve our paper.

---

### Decision · Program_Chairs · 2022-01-20

**Decision:**

Accept (Oral)

**Comment:**

This paper proposes an approach for 8-bit fixed point training of NNs, based on a careful analysis of quantization error in fixed-point methods. They present convincing and thorough empirical results in addition to a detailed analysis providing insights about their method. Reviews for this paper were quite split. One reviewer was a strong advocate, asserting that the paper will have substantial impact in the area, and that the authors’ approach of minimizing quantization error for fixed-point training is of substantial practical interest. Other reviewers were concerned that the proposed method was not novel enough, and that the proposed approach was not practical enough to work in realistic hardware use cases. The authors provided substantial detailed responses addressing the majority of reviewers’ concerns, and after following the discussion in detail I agree with the reviewer advocating for the paper, that the paper presents a practical, novel approach with valuable insights for the field from their analysis and results.

I indicated I am certain about this decision, but I would be ok with the paper being bumped down from oral to poster.